# The global geography of artificial intelligence in life science research

Leo Schmallenbach [1], Till W. Bärnighausen[2,3,4] & Marc J. Lerchenmueller [1,5]

Artificial intelligence (AI) promises to transform medicine, but the geographic concentration of AI expertize may hinder its equitable application. We analyze 397,967 AI life science research publications from 2000 to 2022 and 14.5 million associated citations, creating a global atlas that distinguishes productivity (i.e., publications), quality-adjusted productivity (i.e., publications stratified by field-normalized rankings of publishing outlets), and relevance (i.e., citations). While Asia leads in total publications, Northern America and Europe contribute most of the AI research appearing in high-ranking outlets, generating up to 50% more citations than other regions. At the global level, international collaborations produce more impactful research, but have stagnated relative to national research efforts. Our findings suggest that greater integration of global expertize could help AI deliver on its promise and contribute to better global health.

Artificial intelligence (AI) promises to transform the life sciences and, ultimately, medical care[1]. Broadly defined, AI refers to the ability of a digital computer or computer-controlled robot to perform tasks commonly associated with intelligent beings[2]. In the life sciences, AI is already widely used, for example, when computers analyze large amounts of patient data to aid in initial diagnoses, or when algorithms optimize patient enrollment in clinical trials for drug development[3–5]. The high hopes for the growing use of AI technology are reflected in estimates that the global market for AI-based medical care will grow eightfold by 2027[6].

Against this backdrop, the geography of the AI life science research enterprise, i.e., research that incorporates AI in a life science context, is important for at least three reasons. First, a longstanding line of research has documented that scientific advancement benefits from collaboration[7], especially across borders[8]. Research ideas are rarely confined to national boundaries, the talent needed to conduct research is geographically dispersed, and the challenges of a globalized world require the collaboration of international scientists to derive integrated insights[9,10]. Second, and more specific to AI in the life sciences, geographically concentrated research runs the risk of creating biased data foundations that distort inferences and, possibly, lead to biased medical care[11]. Recent research has already documented biases showing, for

example, that the underrepresentation of ethnicities in training data can lead to distortions in prognosis, diagnosis, and treatment[12,13]. As the AI research agenda in the life sciences rapidly accelerates, fueled by national funding and possibly concomitant interests, questions about effectiveness and equity have grown. Third, AI applications in healthcare promise to deliver high-quality medical care without relying on the expensive and complex machinery traditionally required[14,15]. AI-driven diagnostics and treatment plans can be implemented using more accessible and affordable technologies, such as smartphones and simple medical devices. Such democratization of healthcare technology could enable remote and underserved regions to access advanced medical care that was previously out of reach. These regions must, however, partake in AI-powered life science research to ensure that newly developed technologies meet local needs and to build the capabilities and trust needed for application. In short, the geography of AI research matters for harnessing AI's promises to the benefit of global patient populations.

Existing studies on the geography of AI research, both across scientific disciplines and specific to the life sciences, describe a geographically concentrated enterprise. Studies have shown that China and the United States (US) have come to dominate the AI research system in terms of funding, active scientists and, consequentially, the number of publications[16–18]. A recent meta-analysis at the intersection of general

[1]University of Mannheim, Mannheim, Germany. [2]Heidelberg Institute of Global Health (HIGH), Medical School, Heidelberg University, Heidelberg, Germany. [3]Harvard Center for Population and Development Studies, Harvard University, Cambridge, USA. [4]Africa Health Research Institute (AHRI), Durban, South Africa. [5]Leibniz Center for European Economic Research (ZEW), Mannheim, Germany. ✉e-mail: schmallenbach@uni-mannheim.de

and healthcare-specific AI, which reviewed 288 studies across the disciplines of accounting and management, decision sciences, and health professions, documented a rapidly growing body of AI research, with the US and China contributing the most publications[19]. The study that, to our knowledge, comes closest to our focus on the life sciences, analyzed 3529 scientific AI publications between 2000 and 2021, and again found the US and China to be the most productive geographies based on the number of publications[20]. We provide a summary of our literature review in the Supplementary Material (S1).

We extend this emerging and productivity-focused line of research by analyzing the geography of AI research in the life sciences using three dimensions:

1. Productivity, i.e., publication counts at the country level as well as at the level of world regions, with additional stratification of publications by field of AI application.
2. Quality-adjusted productivity, i.e., publications stratified by field-normalized quality rankings of the publishing outlets.
3. Relevance, i.e., forward citations received by a focal piece of research, additionally stratifying citations into accruing from general research and clinical research.

We apply the three dimensions to a sample of 397,967 AI life science publications and 14.5 million associated citations, creating a multidimensional global atlas spanning over two decades of research (2000–2022).

A detailed sampling protocol, variable descriptions, econometric techniques, and sensitivity analyses are outlined in the Methods. In brief, we use keyword-based text mining and machine learning techniques to identify and classify AI research at the intersection of the life sciences and computer science. We use the standard bibliographic reference for life science research, the PubMed database, to retrieve 374,501 AI-relevant publications from life science journals. To cover computer science, we use the OpenAlex database with its comprehensive indexing and identify 23,466 AI-relevant conference proceedings publications with a life science focus. For constructing our global atlas of AI life science research, we pool the datasets and henceforth use the terms "articles" or "publications" to refer to both journal and conference proceedings publications. To proxy the accuracy of our identification approach, we manually inspect a random sample of 300 articles for AI and life science relevance and test our obtained article coverage against a set of AI special issues in life science journals, obtaining corroborating results. We then stratify the obtained 397,967 AI life science publications by the country of affiliation of the lead author of the articles, i.e., the last author where available, and the first author otherwise, reflecting common authorship norms[21,22].

For the first dimension of our atlas, we analyze the geography of production both at the country level as well as at the level of world regions, according to the six world regions defined by the United Nations: Africa, Asia, Europe, Latin America, Northern America, and Oceania[23]. We also stratify productivity by field of AI life science application, employing the OpenAlex content classification algorithm and keyword-based identification of clinical research. To assess the second dimension, we adjust productivity with a field-normalized approximation for quality, distinguishing articles published in the top three ranked journals and conference proceedings publications for a given field. Finally, we assess the relevance of published research by linking 14.5 million forward citations, distinguishing citations arising from general versus clinical research. To analyze the geography of the first two dimensions (productivity and quality-adjusted productivity), we use descriptive data visualization. To assess the geographic variance in the relevance of the research produced, we use negative binomial regression models. This class of models can accurately estimate the influence of geography, content, and quality of research on relevance (i.e., citations) by also accounting for the skewed distributional properties of citations as the dependent variable.

The three-dimensional assessment provides a nuanced geography of the AI life science research enterprise. Asia leads the global production of AI life science research in absolute terms, with China accounting for over 50% of the region's publications. Examining the content of publications reveals that many countries contribute to core AI research areas in the life sciences (dimension 1). When productivity is adjusted for quality (dimension 2), the regions of Northern America and Europe contribute most publications in high-ranking outlets. We also find that the dimensions of quality and relevance are strongly correlated, with research from Northern America and Europe receiving a substantial citation premium relative to other regions (dimension 3). This citation premium appears to be mostly explained by our approximation of underlying research quality. We complement the three-dimensional assessment of geography by examining international (versus national) collaborations, defined as articles with at least two authors on the author byline who are affiliated with different countries. We present evidence for greater relevance of research conducted in international collaborations as opposed to national collaborations. Despite creating research of greater relevance, the share of international collaborations stagnates and the propensity to collaborate internationally differs between world regions.

## Results

### Dimension one: productivity

We begin our assessment of productivity by documenting an exponential increase in global AI life science publications (Fig. 1), quantified by a 20% annual growth rate since 2010.

Continuing with the first dimension of our atlas, we show a geographical concentration of AI life science research in the US (101,195 articles), followed by China (73,129 articles), together accounting for about 44% of cumulative productivity between 2000-2022 (Fig. 2). Of

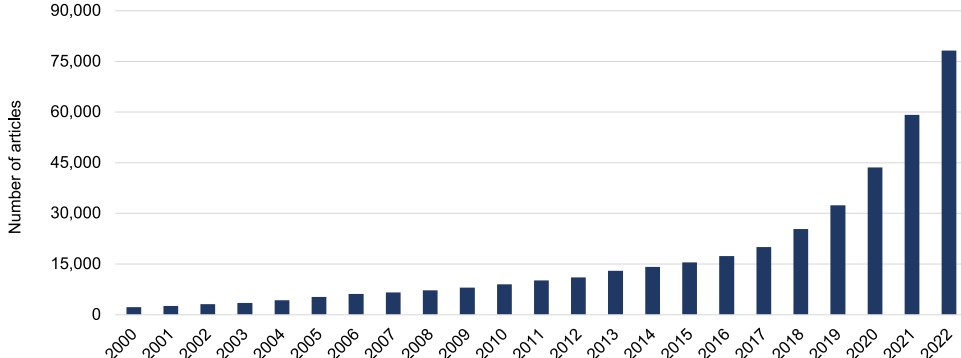

**Fig. 1 | Evolution of the AI research enterprise in the life sciences.** Yearly counts of articles ($n = 397,967$) with AI-related keywords in titles or abstracts from 2000 to 2022. Growth refers to the compound annual growth rate (CAGR 2010–2022). Source data are provided as a Source Data file.

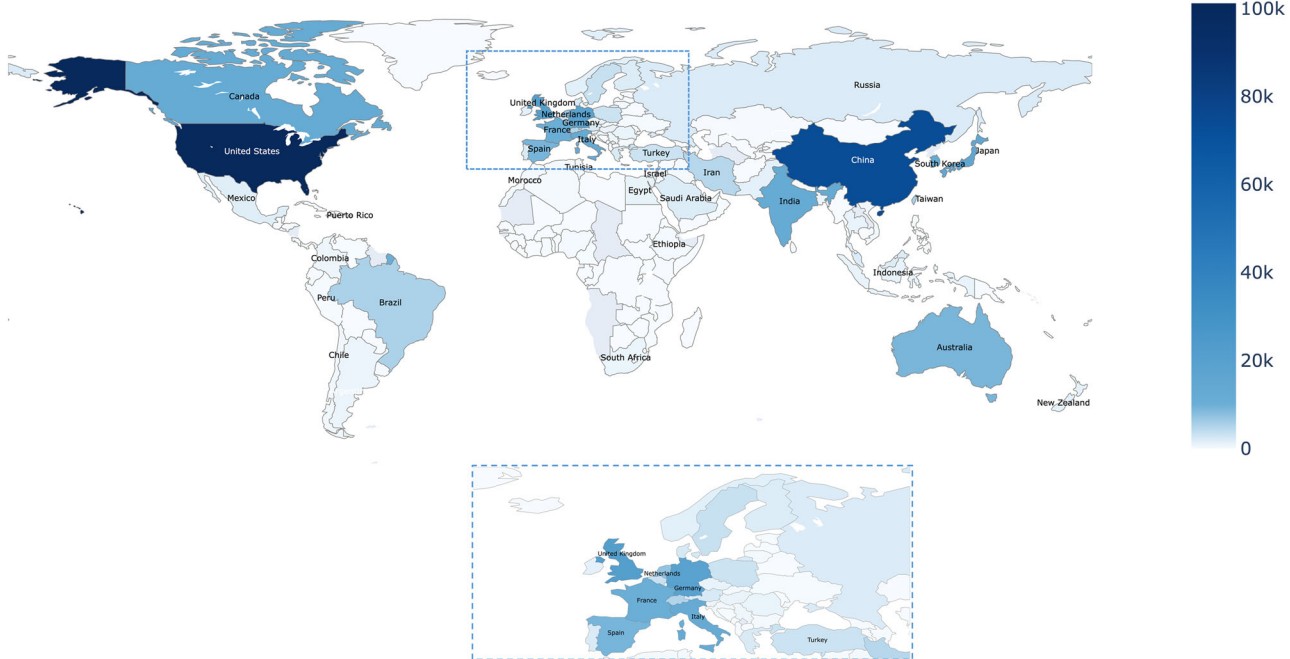

**Fig. 2 | Geography of the AI life science research enterprise in terms of productivity.** Counts of AI-focused life science articles by country, cumulated for the years 2000 to 2022 ($n$ = 397,967). Source data are provided as a Source Data file.

note, 2020 marks the first year in which China has surpassed the US in the number of publications per year in our dataset (see dynamic online graph for details). In terms of cumulative productivity, there is a marked gap between the US, China, and the next tier of countries, which is led by the United Kingdom (21,215 articles), Germany (18,759 articles), Japan (15,263), Canada (12,578 articles), India (12,560 articles), and South Korea (12,264 articles). Select countries, like India, show differences between their productivity in life science journal publications versus computer science conference publications with a life science focus. We provide a table showing all countries' individual productivity statistics in the Supplementary Material S2. While the regions of Asia, Europe, Northern America, and Oceania all tangibly contribute research, countries in Africa and Latin America show moderate-to-low involvement in the AI life science research enterprise. These data underscore two concerns: An almost bipolar geographic concentration of AI research productivity, led by the US and China, while countries from Africa and Latin America remain little involved in AI life science research.

We next consider whether the observed geographic concentration goes in hand with a concentration in research topics and underlying capabilities, which may cater to productivity advantages of some countries over others. In a first step, we assign articles to content categories available from the OpenAlex database. We provide further details on the categorization in the Methods and in the Supplementary Material (S3). We focus our analysis on the 40 most frequent content categories in our dataset, representing, on average, two-thirds of AI life science research across the 40 most productive countries. These 40 countries collectively account for 96% of global productivity in our data. To examine the resulting content-by-country (40 × 40) data matrix, we create a heatmap visualization in Fig. 3. The individual cells of the heatmap contain the share of a country's publications for a specific content category relative to all publications by the same country. This share, expressing nations' research foci, also defines the heatmap's color, with darker shading representing less focus and lighter shading greater focus. The heatmap first indicates that there are many fields that yet stand to gain from further AI applications, indicated by the broad space covered by darker coloring across world

regions. Looking at the most productive AI life science research categories, such as computer vision, computational biology, neuroscience, internal medicine, statistics, radiology, and surgery, there is a global focus rather than geographic specialization. Thus, topic specialization does not appear to be driving the concentration of productivity visible in Fig. 2.

Extending our productivity stratification for content, we assess the extent to which countries generally conduct clinical research with the application of AI. Clinical research is of particular interest because it reflects research with potential applications that more directly benefit human health. To identify clinical research, we rely on a search strategy proposed by Haynes and colleagues[24,25], further described in the Methods. Overall, AI-focused clinical research accounts for about 20% of the articles included in our sample. Figure 4 depicts the geographic distribution across the 30 most productive countries together accounting for 94% of global production of clinical AI research. The primary vertical axis shows the share of a country's clinical research articles relative to all clinical research articles globally (blue bars), while the secondary vertical axis shows the share of a country's clinical research articles relative to all AI life science articles published by that country (orange bars). Comparable to general productivity, we observe the US and China account for about 45% of AI clinical research, with several countries from all world regions, except for Africa and Latin America, contributing tangibly to the clinical AI research enterprise. Consistent with the content analysis presented in Fig. 3, we also find that many countries devote 15–20% of their research efforts to AI clinical research.

### Dimension two: quality-adjusted productivity
Next, we examine whether the geographic concentration we observe in the number of publications is accompanied by a concentration in quality. Scientific progress tends to be driven by research of unusual rather than average quality[26,27], traditionally motivating dedicated examinations of the right-hand tail of the research quality distribution[28].

To adjust for quality with a field-normalized approach, we use external rankings of journals and conferences. For journal articles, we

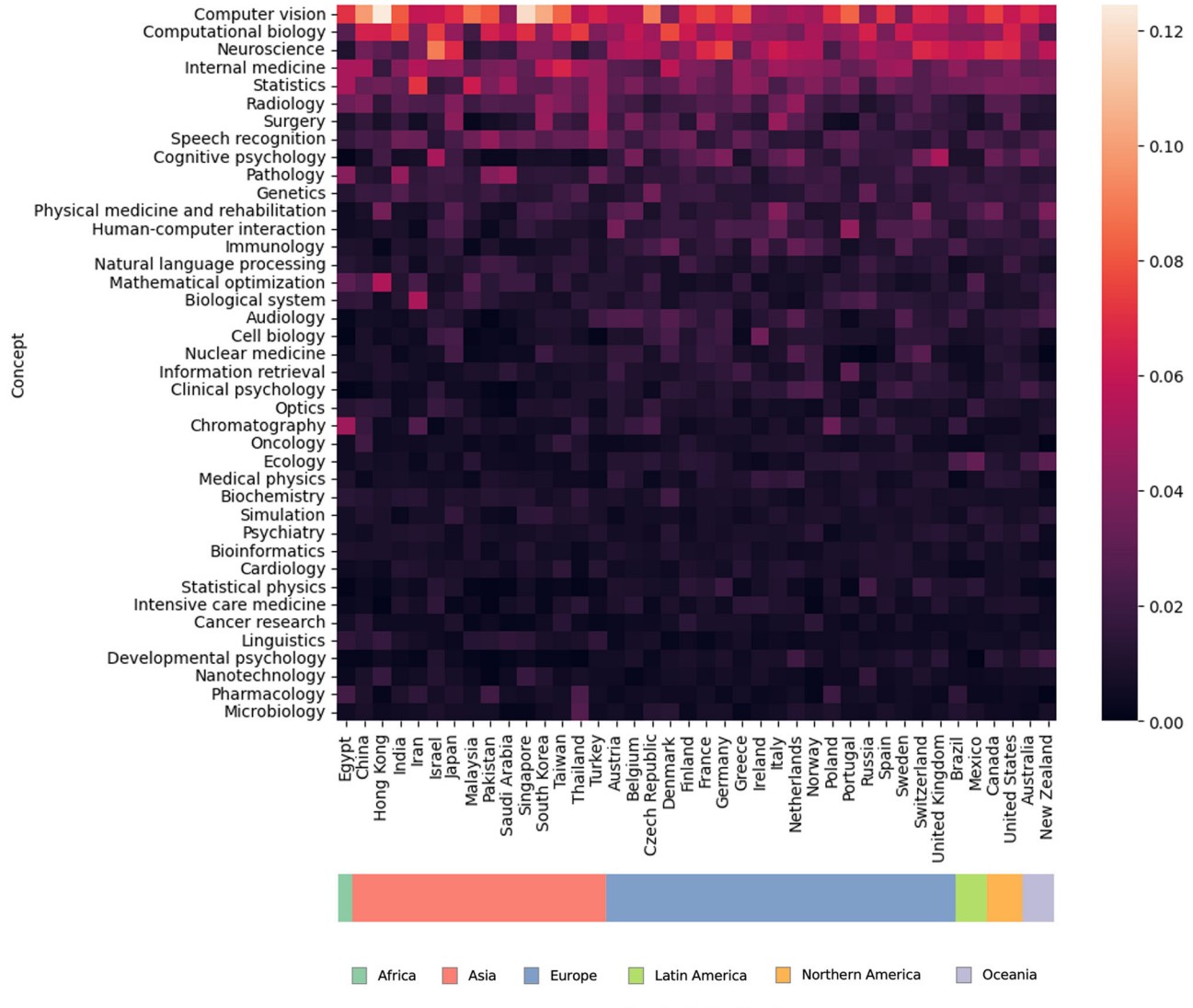

**Fig. 3 | Heatmap of relative country focus with respect to publication topics.** The horizontal axis enlists the 40 most productive countries grouped by geographic region. The vertical axis depicts the underlying publication topics in descending order (computer vision being the most frequently researched topic). The color scheme of the heatmap reflects the percentage share of country-specific productivity for a given publication topic ($n = 397,967$). Source data are provided as a Source Data file.

consider articles published in one of the top three journals within a given journal category according to Clarivate's Journal Citation Report. For conference proceedings, we consider articles published in a proceedings publication of conferences ranked "A*", according to the CORE conference ranking[29]. For journal publications, this approach classifies about 8% of the research as appearing in high quality outlets, and for conference proceedings publications about 6% (S4).

We find that the US, Australia, and several European countries contribute the largest shares of research in high-quality outlets over the period 2000–2022 (Fig. 5). Compared to general productivity, China, and other Asian countries, as well as countries in Latin America rank in the midfield towards the lower-end of the quality-adjusted productivity distribution. Africa, meanwhile, remains largely absent from this mapping due to overall low productivity, including in top-ranked outlets. A notable exception is Kenya, which has international collaborators on two-thirds of its publications placed in high-ranking outlets, while, for example, one-third of South African publications have international collaborators. We discuss the role of internationally collaborated research in a separate section below.

Moving the analysis from the country level to the level of world regions, we seek to examine the consistency with which regions can contribute to AI-focused life science research published in high-ranking outlets. Figure 6 depicts relatively stable proportions of research that distinguish into two groups of regions. On the one hand, there is the group of Northern America, Europe, and Oceania that places consistently about 10% of their published research in high-ranking outlets. On the other hand, there is a group consisting of Asia, Latin America, and Africa, who publish about 5% of papers in these top-ranked outlets. Europe and Asia have shown opposite trends in recent years, with Europe gradually decreasing and Asia gradually increasing their respective shares of publications in high-quality outlets.

**Dimension three: relevance**
To assess the third dimension of the atlas, we examine geographic variance in the relevance of the produced research. We conceptualize relevance as the extent to which focal publications inform (a) scientific progress (scientific relevance) and (b) clinical

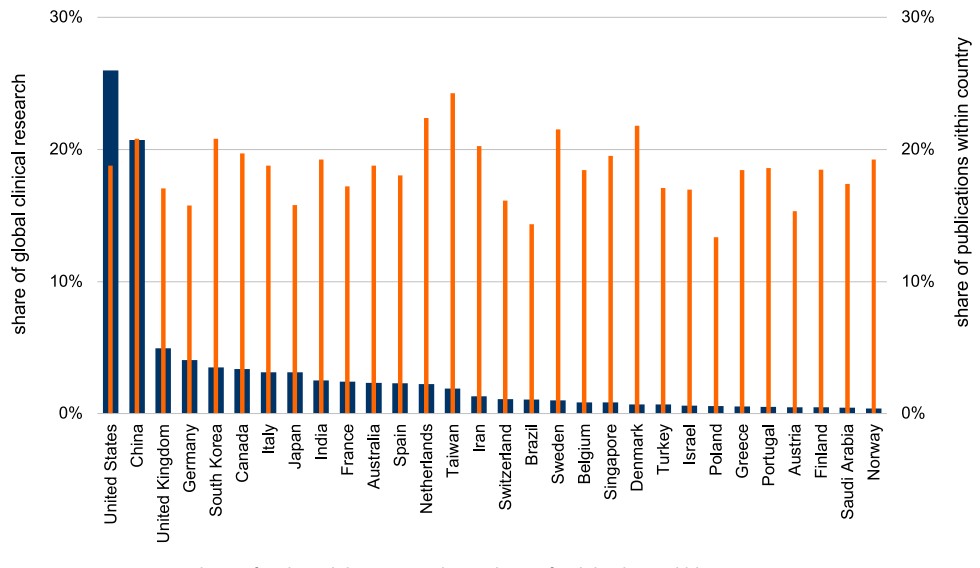

**Fig. 4 | Clinical AI research across countries.** The share of a country's clinical research relative to global clinical research production (primary *y*-axis) and relative to all publications within the same country (secondary *y*-axis) for the 30 most productive countries in terms of clinical articles (*n* = 67,167). Source data are provided as a Source Data file.

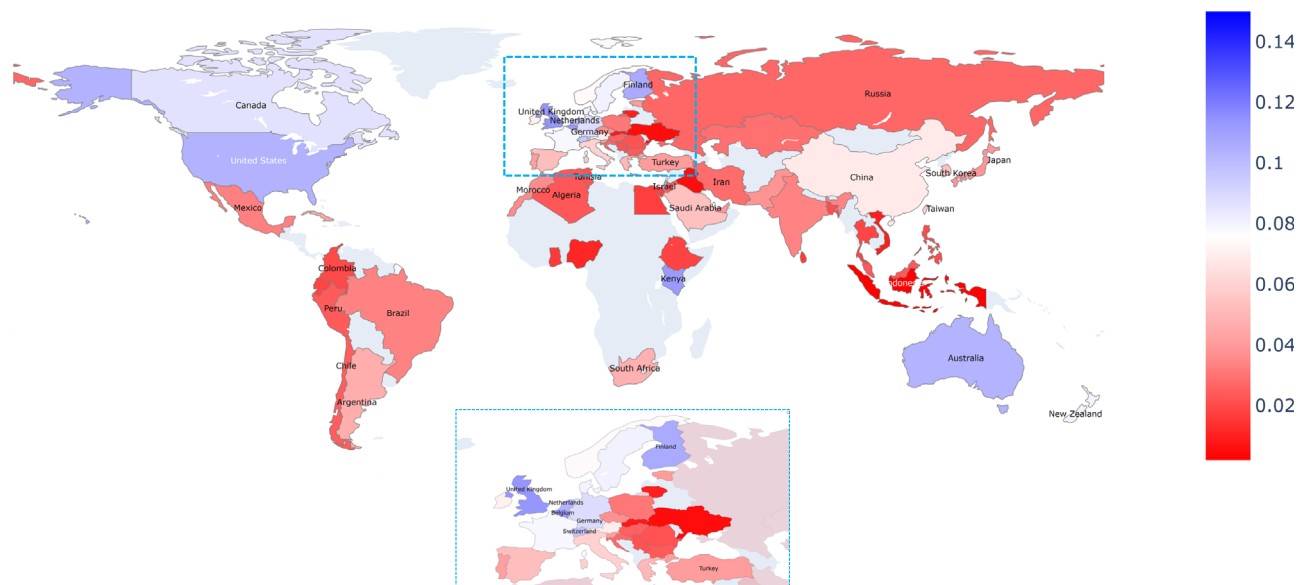

**Fig. 5 | Geography of the AI life science research enterprise in terms of quality-adjusted productivity.** Percentage shares of AI-focused life science articles published in high-ranked outlets by country, cumulated for the years 2000 to 2022 (*n* = 31,837). The analysis is limited to countries with at least 100 publications. Source data are provided as a Source Data file.

application (clinical relevance). We operationalize relevance via forward citations to the AI life science articles in our sample. As econometric model, we employ negative binomial regression models to account for the overdispersion of citation measures. We regress citation counts on dummy variables representing the six geographic regions, setting the most productive region, Asia, as the base category. We control for the publication year to account for the time a given article had to accrue citations. Figure 7 shows incidence rate ratios (IRRs) obtained from the negative binomial regression models. These ratios can be interpreted as percentage changes in the dependent variable, citations, given a one-unit change in the

independent dummy variables, i.e., given the geography of the focal articles across the six world regions.

## Scientific relevance

We assess an article's scientific relevance as the number of forward citations an article receives from general life science research articles. We find that AI-focused life science research produced in the world regions of Africa, Oceania, Europe and Northern America receives about 10% (95% confidence interval (CI) 6%–15%), 26% (95% CI 23%–29%), 20% (95% CI 19%–22%), and 40% (95% CI 38%–42%) more forward citations in general life science articles, respectively, than

research created in Asia (Fig. 7A). Research produced in Latin America, in comparison, receives fewer forward citations than research from Asia.

To adjust for the quality of the underlying research, we next include dummy variables for each journal and conference proceedings outlet in our regression model (i.e., outlet fixed effects). The inclusion of fixed effects adjusts for any geographical variance in research tied to the outlet, including quality ranking and subject matter published. In this adjusted model, with the exception of Africa and Latin America, world regions are no longer statistically different in terms of forward

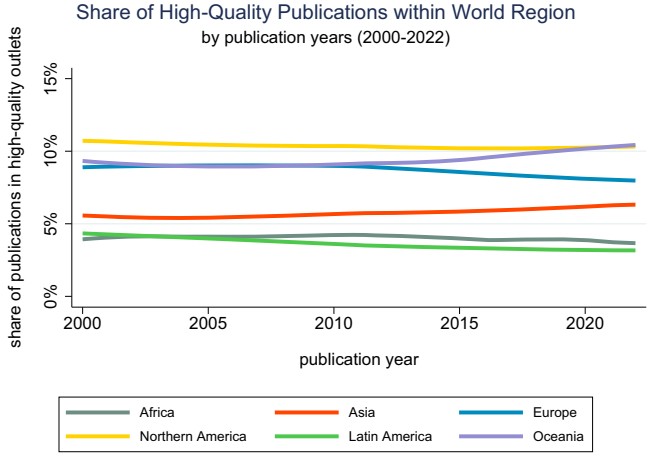

**Fig. 6 | Geography of the AI life science research enterprise in terms of quality-adjusted productivity.** Percentage shares of AI life science articles published in high-ranked outlets by geographic region and per year ($n = 32,010$). Source data are provided as a Source Data file.

citations in downstream life sciences research (Fig. 7D). In other words, the citation differences between geographic regions appear to be largely explained by regional differences in research quality, which is consistent with the geographic variance in research quality shown in Fig. 5.

## Clinical relevance

Ultimately, AI is expected to transform medicine. We therefore seek to analyze the influence of AI life science research on clinically applied research. Figure 7B shows that the regions of Oceania, Europe and Northern America receive a citation premium from downstream clinical research articles (about 13% (95% CI 8%–17%), 26% (95% CI 24%–29%), and 55% (95% CI 52%–57%) respectively), compared to research generated in Asia, analogous to the scientific relevance dimension (Fig. 7A). The greater number of clinical citations to AI life science articles from these three regions again appears to be explained by our approximation of the underlying research quality (Fig. 7C).

Overall, the findings in Fig. 7 indicate that the differences in scientific and clinical relevance are driven by differences in quality rather than geographic bias in citation patterns. In other words, the cumulative knowledge-building process in the AI research enterprise appears to be largely unbiased with respect to the geographic location of the knowledge-creating researchers.

## International collaborations

Lastly, we return to the argument that scientific progress is driven by collaborating on the best ideas, irrespective of the ideas' geography. We analyze international collaborations in our dataset and define articles as international if at least two authors on the author byline are affiliated with institutions from different countries. We focus this analysis on the relevance dimension, because it is the best proxy for

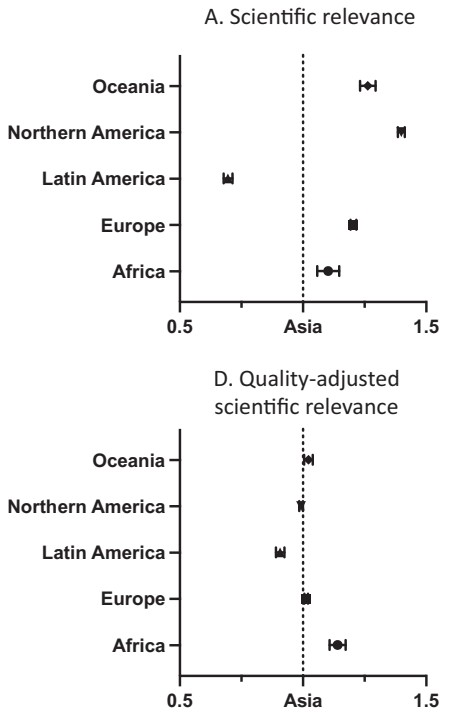
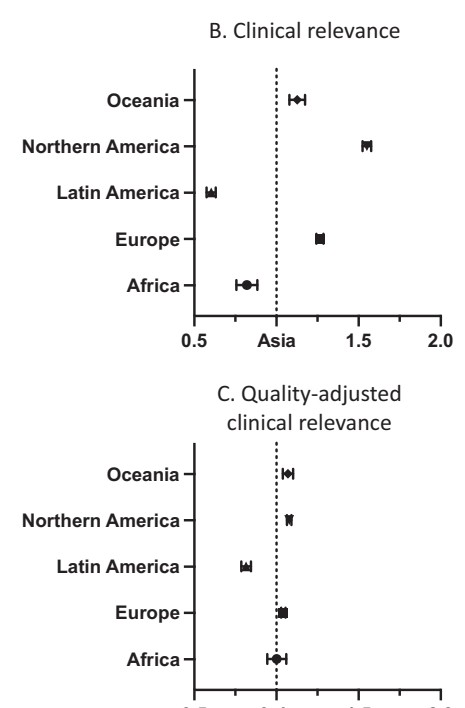

**Fig. 7 | Geography of the relevance of AI life science articles in terms of forward citations in life science research (Figs. 7A, D) and clinical research (Figs. 7B, C).** All panels depict incidence rate ratios (IRRs) with error bars for 95% confidence intervals obtained from negative binomial regressions of citations on dummy variables for the geography of the research, with the most productive region, Asia, serving as the base category. **A** ($n = 397,965$) and **B** ($n = 397,965$) show unadjusted estimates (only accounting for publication year), whereas (**C**) ($n = 393,722$) and (**D**) ($n = 375,033$) also include controls for quality variation across publishing outlets. As publishing outlets with all zero outcomes for the dependent variable (i.e., publishing research that is not cited) get automatically dropped from the analyses with quality controls, the sample sizes are smaller in (**C** and **D**). Source data are provided as a Source Data file.

what kind of research informs the advancement of the global research enterprise.

We again estimate negative binomial regression models with citations as dependent variables and a dummy variable for international collaboration as the core independent variable. In the analysis of a potential citation differential between research from international versus national collaborations, we control for three factors. We include dummy variables for the lead author country to account for regional variance in international collaboration. We control for the number of co-authors because larger author teams are more likely to include a co-author from another country and team size has been shown to correlate with citations[7]. Additionally, we control for the publication year of a focal article to account for the time it had to accrue citations.

We find that articles stemming from international rather than national collaborations receive, on average, 21% (95% CI 20%–22%) more citations by general life science articles and 7% (95% CI 6%–8%) more citations by clinical life science articles (Fig. 8A). Of note, international collaborations also tend to publish 35% more frequently in high-ranking research outlets than national collaborations, on average.

Despite apparent benefits of collaborating across borders, the share of internationally collaborated research is with less than 20% over time relatively low and has come to stagnate in proportion (Fig. 8B). However, the extent to which regions engage in international collaboration varies. Figure 8C shows the share of publications that stem from international collaboration by region of the lead author. While African lead authors coauthor 36% of their publications with at least one collaborator from a different country, Asian lead authors do so for only 16% of their articles. Oceania (32%), Europe (27%), and Latin America (23%) range in between, whereas Northern America also tends to emphasize national over international collaborations (18%).

To further contextualize this cross-regional variance in international collaborations, our final analysis characterizes the dyadic relationships between regions that engage in international collaborations. Figure 9 presents an alluvial diagram to show patterns of international collaboration, including, by construction, only the articles identified as international. We count each occurrence of a difference in geographic location separately and sum international collaborations to the regional level. In other words, if a lead author's country of affiliation is

## International collaborations

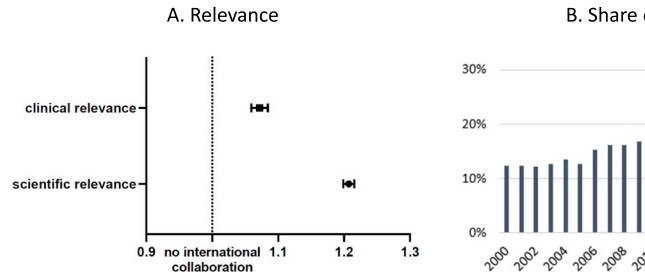
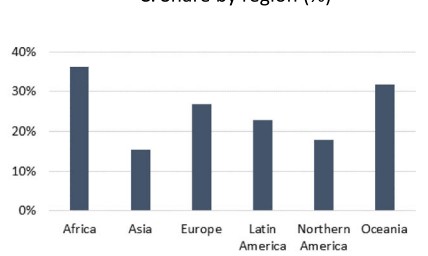

**Fig. 8 | Characteristics of international collaborations.** The effect of international collaboration on scientific and clinical relevance (**A**); share of international collaborations over time (**B**); share of international collaborations by region (**C**). Incidence rate ratios (IRRs) with error bars for 95% confidence intervals obtained from negative binomial regressions of citations ($n = 397,949$) and clinical citations ($n = 397,887$) on a dummy variable for international collaboration, accounting for country of lead author, team size, and publication year (**A**). Percentage share of articles with at least two authors affiliated in different countries ($n = 397,965$) (**B**). Percentage share of articles with at least two authors affiliated in different countries by geographic region ($n = 397,965$) (**C**). Source data are provided as a Source Data file.

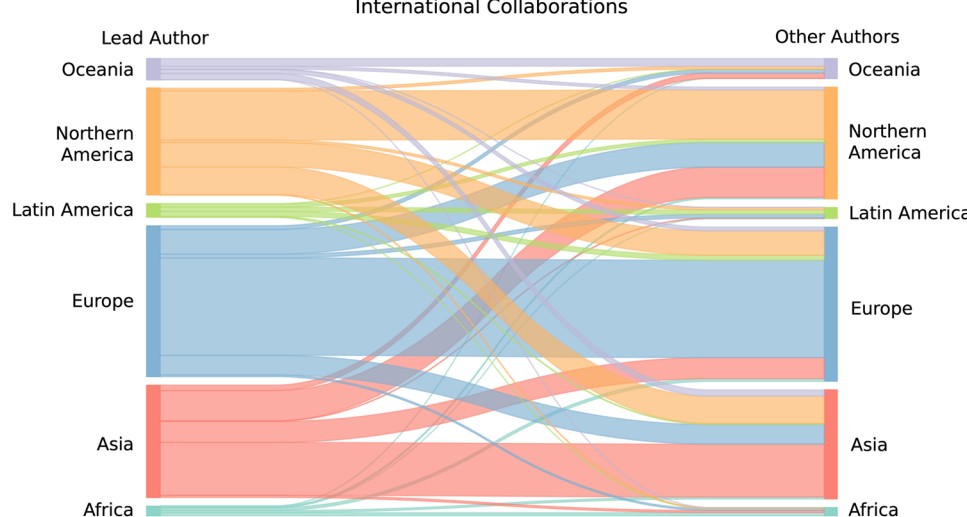

**Fig. 9 | Alluvial diagram of international collaborations.** Number of dyadic collaborations between authors from different countries, aggregated to the regional level. Dyadic collaborations are counted as co-authorships between a publication's lead author (last author or first author otherwise) and any other author on the author byline that is from a different country. Only international dyads are considered ($n = 105,258$ dyads). Source data are provided as a Source Data file.

the US and the lead author collaborates with co-authors from China and Germany, then we depict two lines in the alluvial diagram, one from Northern America to Asia and one from Northern America to Europe. The vertical bars on the left depict the sum of outgoing international collaborations from lead authors affiliated in the respective region, while the right vertical bars depict the incoming collaborations for non-lead authors from the respective region.

Overall, Fig. 9 shows that Europe engages most frequently in international collaborations, both from an outgoing perspective (lead authors) as well as an incoming perspective (other authors), represented in the blue vertical bars on both sides of the diagram. European researchers most frequently collaborate with colleagues from the same geographic region, followed by Northern America. But Europe also appears to play an important role in partnering with African and Latin American researchers. Oceania's international collaborations appear most pronounced with Asia. Africa collaborates frequently with European and Asian researchers. Latin America appears more varied in its international collaboration patterns, but also appears collaborating most frequently with researchers based in Europe. Northern American lead authors tend to mostly co-author with colleagues from the same region, followed by collaborations with Asia and Europe.

## Discussion

Prior research with a focus on productivity has identified the US and China as the main contributors of AI research. In this study, we first test whether this general finding holds true in the specific context of AI applications in life science research and document that the US and China produced almost half of the global AI life science research between 2000 and 2022. Taking a regional perspective, Asia leads global production. We then extend this one-dimensional perspective on research output by considering two additional dimensions: quality-adjusted productivity and relevance. We show that the geography of global AI life science research changes depending on the dimension under consideration. For example, we show that the world regions of Northern America and Europe produce most life science articles published in high-ranking outlets and, alongside Oceania, produce work that most advances the AI life science research enterprise. Meanwhile, the world regions of Latin America and Africa markedly lag as contributors to the AI life science research enterprise. We show that exceptions to this pattern, like select African countries, disproportionally engage in international collaborations to produce research in high-ranking outlets that is also of high relevance.

The productivity-focused analysis of AI research in prior literature has contributed to concerns about national research agendas potentially undermining the effective and equitable advancement of AI research across science fields. In the wake of rising nationalism and protectionism, researchers have come to summarize these bipolar geographic dynamics as a China–US "arms race" in AI[30]. The public discourse feeds this conception. For example, in 2017, China announced a program for the domestic development of AI with the objective of becoming the world's leading AI region by 2030 and has recently underscored its ambitions by pledging a record investment in AI-enabling infrastructure[31]. As AI research uniquely requires large-scale investments, including scaled computing resources, trained human capital, and encompassing data, China successfully accelerates its AI research program[32]. These geopolitical dynamics invigorate the "arms race" perspective on AI research, which is to be appreciated in light of evidence that governments' AI investment can also be politically motivated[33]. Our assessment of productivity mirrors the US-China duopoly perspective also for the life sciences, and it remains subject to further research on how this geographic concentration in productivity influences the advancement of AI in the life sciences and elsewhere longer term.

However, considering quality-adjusted productivity and relevance as two further dimensions of evaluation provides additional, different perspectives. Other world regions, home to many countries, produce AI life science research that is disproportionally used in advancing the AI research enterprise. Research from the regions of Europe, Oceania, and Northern America gets cited at 1.2 to 1.5 times higher rates in general and clinical research when compared to other world regions. This citation premium appears explained by the quality of the underlying research. Some of this quality stratification proxied by the field-specific ranking of the outlets publishing the research may stem from differential access to journals or conferences across geographies. Not all researchers may be equally comfortable publishing their findings in English, generally the standard language of academic communication in international journals, for example. Still, journal and conference proceedings publications remain the central avenue for cumulative knowledge building in the sciences[34]. Overall, our results show that the regions of Northern America, Oceania, and Europe are key regions producing relevant research to advance the AI life science research enterprise.

Geographic differences in research quantity and quality may further hold implications for how scholars model the evolution of the AI research enterprise more generally. Studies predicated on quantitative productivity have mostly equated investments in inputs with superior outputs. For example, cities able to attract the largest number of AI scientists have been found to emerge as the cities accomplishing the largest number of AI publications[16]. Similarly, national research funding has been correlated with publication output[17]. These findings notwithstanding, our study argues and shows that output can be conceptualized along multiple dimensions. While countries in our study each individually operate with a given set of inputs for producing AI research, there is considerable variance in output when comparing quantity to quality and ensuing relevance. As such, we submit that further research is needed that examines the input-output relationship in AI research, in the life sciences and possibly other disciplines, to better understand research trajectories.

Beyond different types of geographic concentrations, we find that international collaborations produce more relevant research than national collaborations. Consistent with previous research highlighting the importance of scientists collaborating across borders[8–10], we find a citation premium of more than 20% for international versus national collaborations, specifically in the AI life sciences context. Despite this apparent importance of internationally conducted AI research for cumulative knowledge building in the life sciences, the rate at which scientists collaborate internationally appears to stagnate. Exceptions to this pattern emerge in select countries that successfully use international collaborations to disproportionally produce research in high-ranking outlets and of high relevance. While the average rate of international collaborations hovers around 20% in our data, countries like Kenya collaborate internationally on over 40% of their publications and even on two-thirds of the publications placed in high-ranked outlets. International collaborations may thus prove instrumental for broadening geographic participation in the AI life science research enterprise.

We also find, however, that the proclivity to international collaboration varies. The most productive world regions of Northern America and Asia team across borders at the lowest rates, while scientists located in Europe collaborate internationally at higher rates and more geographically distributed. Europe collaborating internationally may thus particularly cross-pollinate research conducted in the world regions of Africa and Latin America, which overall create a tangible share of their productivity through international collaborations.

Lastly and importantly, our global atlas shows many world regions remain moderately or little involved in the AI life science research enterprise. Countries in Africa and Latin America account for less than 5% of global AI research in the life sciences. These two world regions are home to more than 25% of the world population and experience more than half of the global disease burden[35]. That is not to say that the existing research does not tackle research questions that are also germane to these regions. In fact, the possibility of scaling AI

applications across world regions may lead to marked benefits for many countries, even if countries are not all involved to a similar extent in the creation of the research. Still, our findings add to a concern that may be especially applicable to the life sciences. The prowess of AI often depends on the data foundation fed to learning-based models. If research remains geographically concentrated, it stands to reason that data foundations evolve in an unbalanced fashion. In turn, the imbalance could lead to biased AI models producing biased recommendations. Patient populations are diverse in terms of gender, race, and ethnicity, as well as other attributes, like socio-demographic status or access to healthcare systems. To mitigate the risk of AI informed medical care being biased towards certain demographics, straddling these different characteristics requires more and accelerated research and building the necessary capabilities and training datasets globally. Parts of the life science community have voiced these concerns, and studies have begun to selectively expose such biases[36]. Our global atlas may be viewed as underscoring the geographic magnitude of these concerns and points to examining the desirability and design of potential countermeasures.

Our study is not without limitations. First, we use the affiliation country of the lead author (last author where available and first author otherwise) to determine the geography of a focal article. Although this approach is in line with characterizing research according to the characteristics of lead authors[21,37], it still focuses on the academic creators of research. Future work may enhance the global atlas of AI life science research by, for example, considering the location of supporting funding institutions or the geography of academy-industry collaborations. Second, we rely on a keyword-based identification of clinical research to distinguish the nature of forward citations. Future research examining the nature and detailed content of clinical studies seems warranted. For example, scholars might more qualitatively examine the kind of clinical research that draws on AI techniques, as well as characterize the medical fields most poised to benefit clinically from AI. Finally, the goal of our study is to provide an atlas of AI life science research. As a corollary, we focused on the "supply side" of research. Bridging this supply-side perspective to a demand-side perspective seems a fruitful research area, addressing questions like what patient populations stand to gain (or lose) from AI advances.

Third, our research, which covers AI applications in the life sciences through 2022, largely misses the very recent surge in studies using large language models (LLMs). These models are poised to have a major impact on a range of biomedical applications, from synthesizing expert literature to improving patient communication and medical education. The rapid development and integration of LLMs highlight the need for ongoing research to better understand their capabilities and ensure equitable benefits. Consideration of the geography of LLM applications is likely critical to addressing access disparities, understanding local implementation challenges, and promoting global health equity.

In conclusion, our study offers a global atlas of AI life science research published between 2000–2022 along three dimensions: productivity, quality-adjusted productivity, and relevance. We show that geographic gravity changes across these dimensions. Overall, the productivity dimension shows Northern America and Asia to dominate, led by the US and China respectively. By contrast, the world regions of Northern America, Oceania, and Europe, with several countries contributing to publications in high-quality outlets, produce research most relevant for advancing the AI life science enterprise. The world regions of Latin America and Africa remain largely absent from the global atlas of AI life science research. Beyond this differentiating geographical view, we show that integrating international collaborations is instrumental for the creation of relevant research. Yet, the internationality of the AI life science research enterprise stagnates. To best advance AI research, concerted international efforts may be warranted.

## Methods

To identify AI-focused life science articles, we take a two-pronged approach that reflects the interdisciplinary nature of this research. On the one hand, we start from the life sciences perspective, turning to the PubMed XML database as the world's most comprehensive inventory of biomedical literature, with more than 35 million articles linked to a range of supporting information. On the other hand, we start from the computer science perspective, turning to articles published in conference proceedings indexed in OpenAlex, a database successor of Microsoft Academic Graph (MAG), containing detailed bibliometric information on more than 250 million scholarly works. Recent research documents OpenAlex to have the widest coverage of academic publications, especially for non-journal publications[38]. We adopt the search strategy established by Baruffaldi et al.[29], which relies on an encompassing keyword search for articles containing AI terms in their title or abstract, and we apply the approach to both data foundations. Baruffaldi and colleagues followed a three-step approach to create a set of query terms for bibliometric databases that accurately retrieve documents focused on AI. In the first step, the authors identified articles published in AI-tagged journals and conference proceedings according to the All Science Journal Classification (ASJC). In the next step, the authors identified keywords listed in these documents and performed a co-occurrence analysis of these keywords based on the titles and abstracts of the AI-tagged documents. Only keywords that appeared at least 100 times and belonged to the top 60% in terms of relevance were kept. Finally, this list of keywords was presented to and approved by a group of AI experts from academia and industry. We provide the final list of 214 AI-related keywords in the Supplementary Material (S5).

For our identification of AI-focused life science research indexed in PubMed, we use these 214 keywords to identify publications containing any of them in either the title or abstract of articles published between 2000 and 2022. In total, we identified 388,633 AI-related life science articles indexed in PubMed. To identify AI-focused life science research in conference proceedings, we first searched for the 214 AI keywords in the title and abstract of 2.4 million documents linked to all 10,794 conferences listed in OpenAlex. In the next step, we apply the content classification embedded in the OpenAlex database to identify AI research that also addresses concepts relevant to the life sciences. OpenAlex tags articles with multiple concepts representing their topical focus, using an automated state-of-the-art machine learning classifier based on titles and abstracts with confidence scores indicating relevance[39]. These scientific concepts are organized hierarchically, with 19 root-level concepts branching into six levels of specific topics. When a lower-level concept is mapped, all its parent concepts are mapped as well, ensuring comprehensive coverage[39,40]. We consider articles that have been assigned at least one of the following four top-level concepts (defined as level 0 in Open Alex) related to life science research: Biology, Chemistry, Medicine, or Psychology. We cross-verify the representativeness of these terms for life science research in our sample of PubMed articles. Here, these four terms represent more than 80% of the indexed life science research. This approach gives us 28,848 conference proceedings publications at the intersection of AI and the life sciences.

We evaluate the accuracy of our approach, summarize the results here, and provide further details in the Supplementary Material (S6). To test precision, we examine a random sample of 150 PubMed articles and 150 conference articles from our final dataset, employing two independent reviewers to rate the PubMed articles for AI focus and the conference articles for life science relevance. In an iterative process, 90% of the PubMed articles were designated as containing a certain type of AI application and 93% of the conference articles as having a life science focus. In addition, we evaluate the comprehensiveness of our approach in identifying AI life science research by looking at the coverage of articles published in AI special issues of life science journals. In

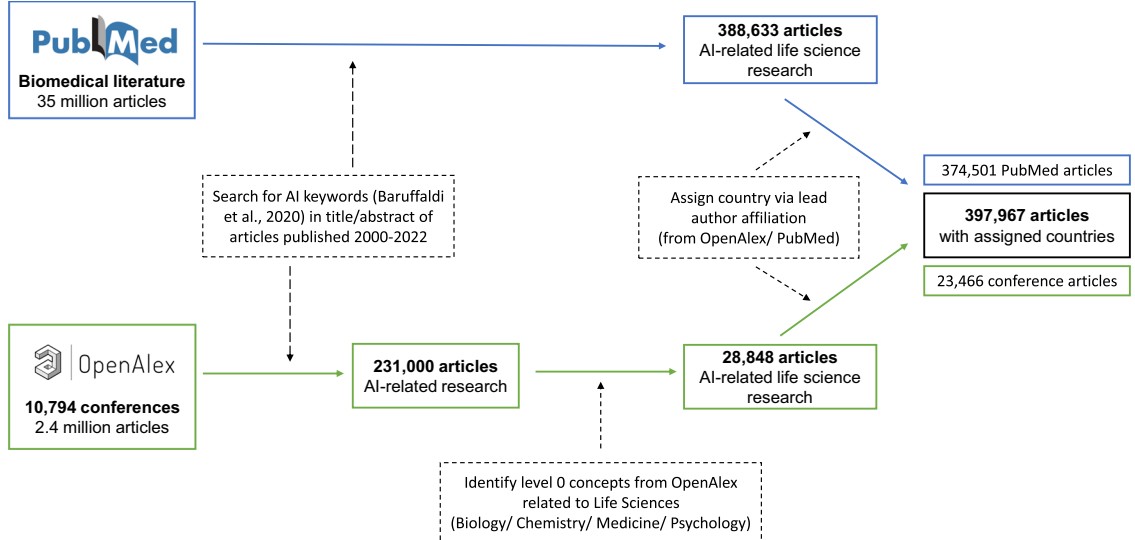

**Fig. 10 | Sample creation.** Overview of sample creation using PubMed and OpenAlex as main databases.

a sample of 15 special issues published in 2022, we find that 92% of the articles published in these special issues are also part of our sample. We also compare the chosen search strategy with a second search strategy for AI-related publications proposed by Liu and colleagues[41]. We find that while the Liu et al. approach is slightly more precise, the approach proposed by Baruffaldi and colleagues yields more than twice as many articles, making it more comprehensive.

We map the 417,481 articles identified with the above search approach to countries using the affiliation of the lead author as recorded in OpenAlex, if available, and otherwise as recorded in PubMed. To link the corpus of PubMed articles to OpenAlex, we leverage unique article identifiers, i.e., PubMed IDs (PMID) and/or Digital Object Identifiers (DOI), and query the OpenAlex application programming interface. To identify the lead author of an article, we use long-established authorship norms[21,22], that reserve the first and last author positions for the lead authors of an article. Since the last author is usually the more senior author, who typically sponsors the necessary research infrastructure (e.g., laboratory and office space), we designate the geographic location of an article based on the country of affiliation associated with the last author when available. Otherwise, we use the affiliation information for the first author. For approximately 90% of the articles in our sample, the affiliation of the first and last author is linked to the same country. We identify a country for 397,967 (95%) articles that make up the final sample of our analysis. We assess the reliability of our country assignment for a random sample of 300 articles with the help of two independent raters and obtain a correct country assignment for 99% of the observations.

Using our final sample, we then map the individual countries to geographic regions according to the United Nations classification: Africa, Asia, Europe, Latin America (including the Caribbean), Northern America, and Oceania[23]. We also collect information on the affiliation of interior authors and use their location to identify international collaborations, which we define as articles where at least one author is affiliated with a country that is different from that of the lead author. Figure 10 summarizes the steps of our sample creation approach.

To enrich our data with information about the content of individual articles, we again make use of the concepts provided in the OpenAlex database. Specifically, we assign each article to the level 1 concepts with the highest relevance score. If this highest scoring concept is a purely AI-related term, we assign the next highest scoring concept to ensure that the assigned concept reflects the context of life science applications. In addition, we create a subset of articles related

to clinical research by performing a keyword search for clinical keywords in the title or abstract[24,25].

We expand the dataset for our quality-adjusted productivity analysis with journal-level information from Clarivate's Journal Citation Report (JCR 2020). We link our data to this report using the unique International Standard Serial Number (ISSN) of the publishing journal. In total, 328,062 (88%) articles of our PubMed corpus were published in a journal indexed by the JCR. Of relevance to our analysis is the journal's rank within the same journal category based on the journal's impact factor. The Pearson correlation between journal impact factors from different vintages of the JCR is generally greater than 0.9, indicating little temporal variance in the scaling of the metric[37]. We consider any publication published in one of the three highest-ranked journals within the same journal category. Because journals can be assigned to multiple categories, we consider the journal category in which a focal journal ranks highest. PubMed articles not published in indexed journals were not considered as Clarivate sets a quality threshold for journal inclusion in the index. Using this approach, we identify 29,510 (8%) articles in our PubMed corpus as being published in high-ranked journals.

We further identify high-quality conference proceedings by making use of an external conference ranking, the so-called CORE ranking, provided by the Computing Research and Education Association of Australasia. CORE provides expert-based assessments of all major conferences in the computing disciplines with information on their research subfield and is a standard resource for ranking computer science conferences[29]. We consider all publications in A*-rated conference proceedings to be of high quality, resulting in 1349 articles (6% of our total sample). We differentiate our analyses between life science and computer science articles in S7.

To characterize the scientific and clinical relevance of an article, we leverage detailed forward citation data from OpenAlex. The database provides not only detailed bibliometrics and metadata but also more than 1 billion citation links between publications. We identify these citation links for the articles in our sample by their PMIDs, DOIs, and OpenAlex IDs. We further distinguish citations into those accruing from any type of publication and those from clinical research only. For this distinction, we again use the keyword-based approach of Haynes and colleagues to identify clinical research[24,25].

## Analysis
We analyze the geography of AI life science research according to the defined three dimensions, namely productivity, quality-adjusted

productivity, and relevance, employing data visualization and regression models. More specifically, we visualize descriptive statistics, including publication counts (Figs. 1 and 2) and geographic percentage shares stratified by content and quality of underlying research (Figs. 3–6). To gauge the scientific and clinical relevance of AI life science research by geographic region, we use negative binomial regression models with the number of citations as the dependent variable and dummy variables for the geographic regions as the main independent variables (Fig. 7). Additionally, we account for publication years using dummy variables in all our regression analyses to normalize for the time an article was at risk of being cited. To estimate the scientific and clinical relevance of AI-related articles conditional on the underlying quality and content, we run additional models with more than 9000 publication outlet fixed effects. These fixed effects (i.e., dummy variables for each outlet) absorb any confounding effects of time-stationary outlet characteristics on citations, including journals and conference proceedings' published content and quality. As journals and conference proceedings with all zero outcomes for the dependent variable (i.e., publishing research that is not cited) get automatically dropped from the fixed effects analyses, the sample sizes are smaller in the corresponding regression models. Our results remain consistent when estimating all models on the smaller samples. We present our results as incidence rate ratios (IRRs) relative to the baseline geographic region (Asia). Finally, we track the share of international collaborations over time as the share of research papers featuring at least two authors with affiliations from different countries (Fig. 8). We estimate the scientific and clinical relevance of these international collaborations in negative binomial regression models, again controlling for publication year dummy variables, the locale of the lead author, and the total number of authors on the author byline. We characterize the geography and direction of international collaborations by considering the country of the lead author as the outgoing country and counting all dyadic connections between the country of the lead author and any other country of the first 10 authors listed on the focal publication (Fig. 9). Importantly, 99% of the publications in our sample list 10 or fewer authors. Analyses are conducted in Stata. Data visualizations are created with Python and Prism.

### Reporting summary

Further information on research design is available in the Nature Portfolio Reporting Summary linked to this article.

## Data availability

The publication data assembled in this study have been deposited in the Figshare database [https://doi.org/10.6084/m9.figshare.24412099]. Source data are provided with this paper.

## Code availability

The computer code to perform the analyses of this study has been deposited in the Figshare database (https://doi.org/10.6084/m9.figshare.24412099).

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

## Acknowledgements

We thank Jennifer Hahn and Luca Caprari for outstanding research support. LS is supported by the Joachim Herz Foundation. MJL received financial support through the FAIR@UMA program of the University of Mannheim. MJL and TWB received financial support in scope of the Helmholtz Information and Data Science School for Health (HIDSS4Health). The publication of this article was partially funded by the University of Mannheim.

## Author contributions

LS, MJL, and TWB devised the original idea. LS and MJL assembled and analyzed the data. LS and MJL wrote the manuscript. TWB edited the manuscript.

## Funding

## Competing interests

MJL is a co-founder and shareholder of AaviGen GmbH, a cardiovascular gene therapy company. The present study is not related to the company. The remaining authors declare no competing interests.
