## [Peer Review File · Nature Communications]

Reviewers' Comments:

Reviewer #1:

Remarks to the Author:

The manuscript characterizes and compares biomedical AI research of different countries, with respects to productivity, quality, etc. For this, the authors collected 175k articles from PubMed and 2.3M citations from iCite. Four dimensions are specifically considered in the comparison: (1) number of publications; (2) IF of published journals; (3) citations; (4) citations in clinical research. The authors found that US and China accounted for about half of all biomedical AI publications during 2000-2022, while North American and European countries published more papers in high-impact journals. In addition, the authors also showed that papers with international collaboration tend to have more citations.

While I appreciate the intent and motivation of this work, I find their data collection and analytic approaches problematic. As such, I have reservations about some results/numbers reported in this study. That said, I do not disagree with their general findings, as none seems really new and unexpected. Please find below my specific comments/concerns:

1. The most important step of this work is to collect all and only relevant biomedical AI papers.

However,

a. Only collecting papers from PubMed introduces reporting biases. PubMed is just the "biomedical view" of biomedical AI research, but as far as I know, a lot of biomedical AI papers that have been published at AI conferences (CVPR, NeurIPS, CHIL, ICHI, ICML, ACL, EMNLP, etc.). The authors must also consider these venues for the corpus collection.

b. The collection methodology is not clearly described. On page 12, The authors simply wrote "Following the strategy ..., we conducted a keyword search for ...", without any further details. I suggest the authors explain how the literature search is conducted more clearly, as this is the key for ensuring reproducibility.

c. The collection methodology has not been rigorously evaluated. Ideally, the collection method should: a) collect only biomedical AI papers; b) recalls all biomedical AI papers. However, the authors have not quantitatively evaluated the collection method, so we do not know whether the paper collection is both (a) precise and (b) comprehensive. For (a), I suggest that the authors randomly sample 100~1,000 articles from the collection, and manually judge whether they are biomedical AI papers. For (b), I suggest that the authors find a corpus of 100~1,000 articles that are certainly biomedical AI papers (e.g., from special issues of some journals), and report how many of these papers are in the collection used in this work.

d. Similarly, the authors should also clearly define what is a "clinical study" and evaluate their "clinical study" classifier when conducting the clinical relevance analysis. The numbers presented in Fig 1B seem extremely low for clinical AI research (e.g., less than 600 papers in total worldwide). For instance, just one journal 'Artificial Intelligence in Medicine' publishes around 150 research articles per year when searched in PubMed.

e. The choice of selecting papers from 2000 and onward seems arbitrary. As a research topic, there are papers on this before 2000 but on the other, it is the last decade where we see a clear surge of interest in such a topic. I'd suggest focusing research papers from 2010.

2. Content-wise, the authors considered four quantitative metrics for analysis, which is mostly straightforward. However, only analyzing the publication count and citation-based metrics seems less informative, and the conclusions are generally not surprising to me. I suggest the authors extend their country-level analysis to the following aspects (both can be extracted from PubMed dump downloads):

a. Publication type, including randomized controlled trials, case reports, reviews, basic research, comments, etc. It can show the different stages of research each country is publishing.

b. Publication topics, such as fairness, different medical disciplines, different AI methods & modalities, etc. It would be interesting to see if different countries have different preferences on these topics.

c. In addition, the authors can also try to normalize each country's statistics by its GDP,

population, R&D investment, and etc.

3. Using JIF as a surrogate for an article's impact is short-sighted. It is problematic because JIF is a rough indication of a journal quality at best, not individual articles. It is puzzling that why a cutoff of 20 is used to consider high impact for biomedical AI research, because this is also unrealistic as there not many specific journals for this research field with an IF of 20 or higher. For instance, papers published in Nature Communications would not be considered as high impact.

4. In terms of the analysis on the internationally collaboration part, it is incomplete when only first and last author were considered, as they tend to come from the same institute in biomedical research papers. Rather, the authors should examine the affiliation of other co-authors in between. It is also known that more authors tend to lead to more citations – this confounding variable should be adjusted in this analysis. Furthermore, for international collaboration, it would also be interesting to show the co-occurrence information to study which countries prefer to collaborate with each other and its temporal trend.

5. The method used to describe Figure 4 should be better explained and described so that readers can fully comprehend its results and findings. For instance, is n in figure 4 the total number of citations? Why is it almost identical for both life science papers vs. clinical AI papers, given there is a significant discrepancy in their quantities in figure 1. Also, what is journal fixed effects? Even after reading the methods part, this is still confusing.

Other specific comments:

1. The first paragraph should start with a clear definition of AI.
2. Fig2A and Fig2B: showing a world map seems nice, but this is not informative: this presentation assumes that the readers are familiar with the country's geographical locations. (I don't know if the authors can label all the African country names on this unannotated map.) The map presentation is also biased towards countries of larger sizes. I suggest that the authors should have an annotated map (and remove Antarctica since it is not analyzed) or use other methods to display the country-level statistics. I would prefer a ranked list of all countries for each analyzed statistic.
3. The authors should clearly describe and compare related studies if this is not the first analysis on such a topic.
4. Icite, icite should be iCite
5. Impact research should be impactful

Reviewer #2:

Remarks to the Author:

This is an interesting analysis that examines the geographic distribution of artificial intelligence research in life science publications. They produce compelling findings about the concentration of AI-based research and citations in the overall literature in the US and China, but a relatively higher distribution of research in other research when accounting for impact factor of journals that publish AI research. Furthermore, they show that international research is more heavily cited in higher-impact venues than national teams. I have several comments:

Introduction:

(1) The first 3 paragraphs could be shortened considerably. An extensive discussion of the possibilities of AI and need for representative training are not needed to get into the crux of the matter, which starts in line 87, that geographic concentration of AI research may not inform care for patients outside of that geography

Methods

(2) Can the authors better define what they consider "life science" vs. "clinical" research? Does

clinical research pertain to phase I, II, III trials? A better definition here would help

(3) The authors need to acknowledge that their classification of "AI-based research" is prone to error.

a. First, they only identify AI from journal titles or abstracts, which may under-identify AI methods that are key to the paper but only mentioned in the methods.

b. Second, AI could apply to many things – methodologies used in a retrospective analyses, derivations of a model, prospective validation of AI. Any differentiation between TYPES of research (beyond just "life science" vs. "clinical") would be very welcome

(4) Did the exclusion of "neural network" articles only apply to biological neural networks

(5) Can the authors cite any statistics that the OpenAlex interface accurately classifies authors? What happens if there is an author with the same name as another author?

(6) It would be interesting to reproduce analyses of AI based research on a per capita basis, e.g. per capita of number of research or per capita of a country's population. It is somewhat expected that countries with higher populations have more AI articles, just like they would have more non-AI articles.

(7) Did the authors collect academic vs. industry-sponsored research? Industry-sponsored research is mentioned in a few places in the manuscript but does not appear to be measured.

Results

(8) How do the authors interpret the lower quality found in Oceania vs. the strong citation premium found?

Discussion

(9) In the discussion, the statements about inadequate access to AI advancements may be premature. Indeed, taking examples from drug discovery, most clinical trials of any cancer agent , for example, are performed in North American and European populations; however, these innovations could be distributed in other populations. One of the potentials of AI, indeed, is the potential for AI to be distributed more widely than traditional clinical innovations. I believe this article is important enough discussing the geographic underrepresentation of authors in AI research and does not need to stray into patient access to technologies.

(10) A key limitation is that the authors comment on geographic characteristics of first and last author, but not on the geographic characteristics of the patients studied in the trial. Where there is a correlation there (and the authors may do well to illustrate this correlation from a subsample of their articles), it is possible that North American authors can publish clinical trials of agents from African or Asian nations, often with authors from these nations included as middle authors. This limitation should be made clear.

POINT-BY-POINT RESPONSE

We would like to thank the Reviewers and Editors for the opportunity of revising our manuscript and offering a set of additional analyses informed by constructive feedback. In the following, we submit a comprehensive point-by-point response that addresses all points raised. We have included the Reviewer comments in **black** and our response to the comments in **blue**.

REVIEWER COMMENTS

Reviewer #1 (Remarks to the Author):

The manuscript characterizes and compares biomedical AI research of different countries, with respects to productivity, quality, etc. For this, the authors collected 175k articles from PubMed and 2.3M citations from iCite. Four dimensions are specifically considered in the comparison: (1) number of publications; (2) IF of published journals; (3) citations; (4) citations in clinical research. The authors found that US and China accounted for about half of all biomedical AI publications during 2000-2022, while North American and European countries published more papers in high-impact journals. In addition, the authors also showed that papers with international collaboration tend to have more citations.

While I appreciate the intent and motivation of this work, I find their data collection and analytic approaches problematic. As such, I have reservations about some results/numbers reported in this study. That said, I do not disagree with their general findings, as none seems really new and unexpected. Please find below my specific comments/concerns:

1. The most important step of this work is to collect all and only relevant biomedical AI papers. However, a. Only collecting papers from PubMed introduces reporting biases. PubMed is just the “biomedical view” of biomedical AI research, but as far as I know, a lot of biomedical AI papers that have been published at AI conferences (CVPR, NeurIPS, CHIL, ICHI, ICML, ACL, EMNLP, etc.). The authors must also consider these venues for the corpus collection.

1. Thank you for drawing our attention to conferences as potential source for AI life science research beyond *PubMed*. We concur with you that the aim of our study is to identify all relevant life science AI papers. We thus would like to emphasize that conference publications as additional source should align with the life science research focus adopted by our study. Accordingly, we have collected articles from the set of suggested conferences and analyzed the extent to which the content aligns with life science research. We consider the life sciences as the “branch of science concerned with the study of living organisms”, adopting the concrete scope from the National Library of Medicine’s description on the coverage of the *PubMed* database, our core corpus of academic publications for analyses.

To retrieve documents for the suggested conferences, we turned to the *OpenAlex* database (<https://openalex.org>). In brief, *OpenAlex* is based on Microsoft Academic Graph (MAG) and represents a core large-scale catalog for the global research system^{1, 2}. Importantly, the database connects identified scholarly entities across multiple datasets, including conference publications.

Table R1.1. below summarizes the number of obtained articles by conference scheme for the time window of our study (2000–2022). Apart from the “Conference on Health, Inference and Learning”, all the suggested conferences were covered in *OpenAlex*.

Conferences 2000-2022	# Documents
CVPR	4.076
NIPS	8.630
ICHI	142
ICML	5.111
ACL	6.170
EMNLP	2.243
Total	26.372

R1.1. Conference documents 2000–2022

Next, we looked at the thematic focus of these articles by analyzing “concepts”, that is keywords *OpenAlex* assigns to each article based on text analysis. To elaborate, *OpenAlex* links every work in its database with one or multiple concepts together with a score that reflects the degree to which the given concept matches the article’s content. *OpenAlex* defines this score as the strength of the connection between the work and this concept – the higher the score, the more discriminant the concept for the article’s content. This score is produced by Amazon Web Services (AWS) Sagemaker, using a machine-learning model with a minimum score needed as quality threshold for assigning a concept to an article. Further details are provided under the header “concept” in the *OpenAlex* documentation (<https://docs.openalex.org/api-entities/works/work-object>).

For determining the focal content of a given conference article, we rely on the assigned concept with the highest score. Given the thematic focus of the suggested conferences, we find that the concept “computer science” generally gets assigned the highest score across documents. While this finding corroborates the concepts provided by *OpenAlex*, we decided to associate the next highest score to an article when computer science scored highest. This approach allowed us to designate each article to one main research theme other than computer science (if available). To compare the thematic focus of the conference articles to our sample obtained from *PubMed*, we apply the same machine learning scoring as used in *OpenAlex* to a random draw of 1,000 articles from our *PubMed* sample. This analysis allows us to verify whether the conference and the life science articles indexed in *PubMed* share a life science research focus, besides a common link to AI.

Main concepts of conference publications vs. PubMed sample

R1.2. Topical focus of conference publications vs. PubMed Sample

As shown in R1.2., the topical focus of the conference publication and PubMed is tangibly different. In particular, the conference publications appear more focused on solely Computer Science (~20% are assigned the concept Computer Science only) and foundations thereof (~50% Mathematics). By contrast, the *PubMed* sample has an emphasis on biomedical research (~35% Medicine, ~15% Biology) and related disciplines (e.g., ~10% Psychology; 7% Chemistry), all part of the life sciences. Observing these differences in research foci, we submit that including the conference documents introduces an undesired thematic deviation. We would also like to point out that even if we were to include the relatively low number of biomedical AI research papers that one could identify among the conference publications, this would likely not affect our main analyses and conclusions because of the small numbers compared to the overall size of our sample.

Reflecting your comment and the ensuing analysis (R1.2.), we have now emphasized the focus of our study being the (biomedical) life sciences in the main manuscript, and copy an illustrative text passage from the start of the Method section below:

“To identify AI-focused life science articles, we turned to the PubMed XML database. PubMed is the most comprehensive inventory of biomedical literature worldwide, comprising more than 35 million articles connected to a range of auxiliary information.”

- b. The collection methodology is not clearly described. On page 12, The authors simply wrote “Following the strategy ..., we conducted a keyword search for ...”, without any further details. I suggest the authors explain how the literature search is conducted more clearly, as this is the key for ensuring reproducibility.
2. We have elaborated on our search strategy in the revised Method section. First, we describe a state-of-the-art search strategy in bibliometric corpora for articles with an AI focus, drawing on the peer-reviewed approach proposed by Liu et al. (2021)³. We also describe in detail the retrieval approach using the *PubMed* database as well as our approach for complementing *PubMed*-based information with additional information from the

OpenAlex database; the latter being important for supplementing location information of authors. Finally, we have added an additional Figure to the main manuscript (Fig.7 pasted as figure R1.3. below) that visually summarizes the sample construction.

* List of 10 Keywords:

"artificial intelligen**"	"deep learning"
"neural net**"	"reinforcement learning"
"machine* learning"	"learning algorithm**"
"expert system**"	"*supervised learning"
"natural language processing"	"intelligent agent**"

R1.3. Sample construction diagram

c. The collection methodology has not been rigorously evaluated. Ideally, the collection method should: a) collect only biomedical AI papers; b) recalls all biomedical AI papers. However, the authors have not quantitatively evaluated the collection method, so we do not know whether the paper collection is both (a) precise and (b) comprehensive. For (a), I suggest that the authors randomly sample 100~1,000 articles from the collection, and manually judge whether they are biomedical AI papers. For (b), I suggest that the authors find a corpus of 100~1,000 articles that are certainly biomedical AI papers (e.g., from special issues of some journals), and report how many of these papers are in the collection used in this work.

3. Thank you for these constructive suggestions. We now offer both precision and recall metrics in the method section of the main manuscript and summarize the metrics below.

To evaluate precision, we took a random sample of 100 articles from our dataset and hired two independent raters to categorize the documents into having an AI life science focus (one) versus not (zero). The observed agreement among raters was 94%.

All papers of the 100 randomly sampled papers were evaluated as having an AI focus. However, for 7 out of the 100 articles, the raters were unsure regarding the article's substantive connection to the life sciences. We then jointly inspected these seven papers, for which we provide a summarizing table below (R1.4). In summary:

- Item 15 (out of the random sample of 100 papers reviewed) is evaluated as foremost a mechanical engineering application and was thus evaluated to reduce precision
- Item 23 is a generic method paper on AI neural networks, which was conservatively evaluated to reduce precision
- Item 37 applies AI to scene recognition, a task relevant in e.g. medical imaging and diagnostics; *the article was classified as increasing precision*
- Item 53 researches AI methods (e.g., DBSCAN) that can be used in epidemiological models of disease spread, for instance; *the article was classified as increasing precision*
- Item 71 researches scene recognition (see item 37), specifically image-denoising, which has become key in medical image analyses; *the article was classified as increasing precision*
- Item 74 researches AI-based image restoration, a critical tool in improving the quality of tomography imaging, for example; *the article was classified as increasing precision*

- Item 77 is evaluated as foremost an information engineering application and was thus evaluated to reduce precision

Item	AI Biomedical	Title	Country	PMID
15	No	pressure prediction model based on artificial neural network optimized by genetic algorithm and its application in quasi-static calibration of piezoelectric high-pressure sensor.	China	28040905
23	No	dynamic event-based state estimation for delayed artificial neural networks with multiplicative noises: a gain-scheduled approach.	China	32916602
37	No	simultaneous semantic segmentation and depth completion with constraint of boundary.	China	31979249
53	No	deepcompnet: a novel neural net model compression architecture.	India	35242176
71	No	optimal combination of image denoisers.	United States	30869617
74	No	self-organized operational neural networks for severe image restoration problems.	Finland	33401226
77	No	support vector echo-state machine for chaotic time-series prediction.	China	17385625

R1.4. Seven candidate papers for reducing precision (out of a random sample of 100 papers).

After reviewing the set of seven papers, the raters and authors jointly designated four papers as AI-focused research with a sufficient connection to the life sciences. In all, we thus obtain 97 (93 + 4) correctly identified papers for our intended research from a sample of 100 randomly selected papers, indicating precision of 97%.

To evaluate recall, we followed your suggestion and identified special issues via *PubMed*. Specifically, we searched for editorials published in 2022 containing the keywords “special issue” and “artificial intelligence”, obtaining nine special issues with a total of 148 papers. In all, 138 of 148 papers were also included in our dataset, indicating a recall rate of 93%.

While there is some recall variance by special issue (with a minimum of 85% at the issue level), the limited variance leads to an average recall across special issues of again 93%. Table R1.5. provides an overview of the special issues and the corresponding recall rates.

A: Journal	B: URL of special Issue	C: # articles in special issue	D: # articles also in dataset	E: Recall (D/C)
Diagnostics	https://www.mdpi.com/journal/diagnostics/special_issues/Dentistry_Oral_Health	15	15	100.00%
Diagnostics	https://www.mdpi.com/journal/diagnostics/special_issues/AI_Cardiopulmonary	14	12	85.71%
Molecular Sciences	https://www.mdpi.com/journal/ijms/special_issues/Learning_Bioinformatics	22	20	90.91%
Skeletal Radiology	https://link.springer.com/collections/fgcagaafg	28	28	100.00%
Veterinary Radiology & Ultrasound	https://onlinelibrary.wiley.com/toc/17408261/2022/63/S1	12	11	91.67%
NMR in Biomedicine	https://analyticalsciencejournals.onlinelibrary.wiley.com/toc/10991492/2022/35/4	13	11	84.62%
Diagnostics	https://www.mdpi.com/journal/diagnostics/special_issues/Computer-Aided_Diagnosis	16	16	100.00%
Radiology: Artificial Intelligence	https://pubs.rsna.org/toc/ai/4/2	19	16	84.21%
Multimedia System	https://link.springer.com/collections/cicahijidd	9	9	100.00%
Total		148	138	
Recall (overall)			93.24%	
Recall (average)				93.01%

R1.5. Nine special issues and recalled publications.

Upon manual inspection of the 10 non-recalled articles, we find that five articles not recalled include AI-identifying keywords in the main text but not in the title and abstract, while two non-recalled articles use more generic terms like “algorithm” thereby evading our applied keyword identification. The remaining three articles were commentaries without dedicated abstracts that reduce the likelihood of keyword-based recalls merely from titles. These reasons arguably point to a lower relevance of the critical AI concepts we seek to identify, either with respect to the article (when the keyword is not included in the abstract) or with respect to the research itself (i.e., generic application of algorithms).

d. Similarly, the authors should also clearly define what is a “clinical study” and evaluate their “clinical study” classifier when conducting the clinical relevance analysis. The numbers presented in Fig 1B seem extremely low for clinical AI research (e.g., less than 600 papers in total worldwide). For instance, just one journal ‘Artificial Intelligence in Medicine’ publishes around 150 research articles per year when searched in PubMed.

- We apologize for not having provided a clearer definition of the clinical study classifier. In our original submission, we drew on the clinical classification employed by Hutchins and colleagues (2019)⁴ that underpins the clinical article indicator in iCite (<https://icite.od.nih.gov>). This classification approach is very specific, essentially classifying articles as clinical when the underlying article is a clinical trial. We paste the exact *PubMed* query from the referenced article below that the authors had used to classify clinical articles. We also include this information in the supplementary material (S5). This very specific classification contributes to the low number of “clinical articles”, yet seemed reasonable for testing specifically the extent to which AI research informs clinical research in an applied sense.

PubMed query: (("clinical trial"[Publication Type] OR "clinical trial, phase i"[Publication Type] OR "clinical trial, phase ii"[Publication Type] OR "clinical trial, phase iii"[Publication Type] OR "clinical trial, phase iv"[Publication Type]) OR "clinical study"[Publication Type]).⁴

Your comment further encouraged us to expand our analysis of what may constitute a clinically-relevant article. We orient ourselves using the definition for clinical research provided by Ioannidis (2016)⁵, according to which “clinical research is meant to cover all types of investigation that address questions on the treatment,

prevention, diagnosis/screening, or prognosis of disease". To operationalize this definition, we make use of the MeSH (Medical Subject Header) thesaurus to identify all articles in our dataset that get assigned by the National Library of Medicine a disease-signifying MeSH term from the C-branch of the MeSH tree structure (<https://www.nlm.nih.gov/oet/ed/pubmed/mesh/mod01/04-100.html>). We identify about ~43,000 articles in the broader dataset of ~175,000 articles that may be designated as clinically relevant according to this broader definition (i.e., 24,6% of our sample). We offer a table of the most frequent disease MeSH terms in the supplementary material (S5).

We can now compare how sensitive the geography of clinical research is to the operationalization of clinical research. R1.6. lists the 15 most productive countries in terms of AI-focused clinical research, together accounting for more than 80% of clinically relevant articles. The ranking of countries and their relative shares of total clinical research productivity are similar for both C-branch MeSH terms (left) as well as iCite definition (right). According to both clinical designations, the US are ahead of other countries, followed by China and the UK. We now offer this sensitivity analysis also in our supplementary material (S5).

	C-branch MeSH term		Clinical (iCite)	
	absolute	relative	absolute	relative
United States	12.495	29%	730	31%
China	7.387	17%	315	13%
United Kingdom	2.249	5%	165	7%
Germany	1.720	4%	147	6%
Japan	1.527	4%	101	4%
South Korea	1.462	3%	84	4%
Italy	1.412	3%	76	3%
Canada	1.397	3%	71	3%
India	1.281	3%	16	1%
France	934	2%	55	2%
Australia	914	2%	48	2%
Spain	877	2%	56	2%
Netherlands	839	2%	73	3%
Taiwan	793	2%	36	2%
Switzerland	491	1%	27	1%
Top 15 countries	35.778	83%	2.000	85%
All countries	43.265	100%	2.345	100%

R1.6. Clinically-relevant research according to C-branch MeSH terms (left) and clinical research indicator from iCite (right), stratified by country

We draw two conclusions from our response to your comment. First, we exclude the clinical subsample from our original Figure 1 in the main text, as a depiction of longitudinal growth in clinical research necessitates choosing among two possible designation approaches that both produce the same finding of exponential growth in AI clinical research. Second, we include the information contained in R1.6. as an additional exhibit in the main manuscript (Fig 2C) to show the difference in geography between general and clinically relevant AI life science research.

- e. The choice of selecting papers from 2000 and onward seems arbitrary. As a research topic, there are papers on this before 2000 but on the other, it is the last decade where we see a clear surge of interest in such a topic. I'd suggest focusing research papers from 2010.
5. We apologize for not having been clearer about our considerations for analyzing articles published from 2000 onwards. Apart from the (subjective) conceptual appeal of covering two decades of research, there are technical reasons related to the XML PubMed database that guided our choice of time frame. Specifically, the National Library of Medicine (NLM) converted over 10 million Medline records starting with the production year 2000 to XML from the NLM legacy ELHILL system. In this process, the NLM also increased the maximum length of abstracts for records created after 2000 to 10,000 characters. Abstracts were truncated for articles indexed prior to the year 2000. Having maximum abstract length is important for our mining of titles and abstracts for AI keywords. And lastly, the PubMed XML began with journal issues published in 2000 to record all authors part of an author byline. Taken together, we deemed the year 2000 as the logical starting point for our analyses, ensuring data completeness while allowing for the greatest temporal coverage.
2. Content-wise, the authors considered four quantitative metrics for analysis, which is mostly straightforward. However, only analyzing the publication count and citation-based metrics seems less informative, and the conclusions are generally not surprising to me. I suggest the authors extend their country-level analysis to the following aspects (both can be extracted from PubMed dump downloads):

a. Publication type, including randomized controlled trials, case reports, reviews, basic research, comments, etc. It can show the different stages of research each country is publishing.

6. Thank you for this set of constructive suggestions. We first retrieved the publication types from *PubMed*. Since articles may be assigned several publication types (e.g., journal article and clinical trial), we considered all type-assignments per article. We then created a heatmap depicting the relative contribution of each article type to the overall productivity per country. For example, the United States produce 55,629 papers, of which 81% get assigned “Journal Article” as publication type. Across the 30 most productive countries, accounting for 93% of the ~175,000 articles we identified, on average 83.75% of countries’ publications are classified as “Journal Articles”. Variance with respect to publication type is also limited, with 75.5% being the lowest share of the category “Journal Article” within country. Journal articles and Reviews account, on average, for 91% of AI-related life science production. As such, the heatmap for publication type (R1.7. below) shows that almost all countries have most of their productivity published in form of journal articles. We reference this information in the Discussion of the main manuscript (also pasted below) and provide further details on the analysis in the supplementary material (**S6**).

“Journal publications remain the central avenue for cumulative knowledge building in the sciences. Our analyses underscore this claim in the context of AI, showing that over 80% of countries’ AI-focused life science research gets published in form of journal articles, an analysis we include in the supplement (S6).”

b. Publication topics, such as fairness, different medical disciplines, different AI methods & modalities, etc. It would be interesting to see if different countries have different preferences on these topics.

7. Creating a heatmap of publication topics also proved an informative analysis. To create this heatmap, we first assigned articles to journal categories available from the Web of Science (Clarivate) based on substance matter published in the respective journals. For example, the journal “Circulation” would get assigned to the journal category “Cardiac and Cardiovascular Systems”, whereas “Nature” would be designated as “Multidisciplinary Sciences”. We include Clarivate’s definitions of the journal categories in the supplementary material (**S4**). We focus the depiction of the heatmap to the 30 most frequent journal categories representing, on average, 74% (standard deviation 4%) of the productivity across the 30 most productive countries in our dataset. Again, these 30 countries produce 93% of the ~175,000 articles in our dataset.

Using this data, we now include R1.8. as a main exhibit in our manuscript (Figure 2B) and describe the key findings also in brief here. The vertical axis of R1.8. ranks the 30 fields according to their relative share in productivity across the countries listed on the horizontal axis. The heatmap indicates that there is an emphasis on the first ranked 10 fields across countries and world regions. As to more regional research foci, there are a few that may be gleaned from the heatmap, for example, Radiology, Nuclear Medicine and Medical Imaging in Western Europe. Certain areas of research may also stand out at the country level, e.g., Biochemical Research Methods for Israel.

Overall, the heatmap elicits global productivity concentration in ten major fields of research with many fields beyond still standing to gain from AI application in the future.

R1.7. and R1.8. Heatmaps of relative country-focus with respect to publication types (R1.7.) and publication topics (R1.8). The horizontal axes enlist the top 30 productive countries grouped by continent. The vertical axes depict the underlying publication type (R1.7.) and publication topics (R1.8.), respectively. The color scheme of the heatmaps depict the percentage share of country-specific productivity for a given publication type (R1.7.) and a given publication topic (R1.8.), respectively.

c. In addition, the authors can also try to normalize each country's statistics by its GDP, population, R&D investment, and etc.

8. We agree that scientific production is correlated with measures of investment into science, e.g., GDP, %R&D investment of GDP, scientific workforce etc. That said, our manuscript seeks to create a global atlas of contributions to the AI life science research enterprise across three dimensions, assessing the geography of research with respect to publications (dimension 1), underlying content and quality (dimension 2) and, ultimately, relevance (dimension 3). We submit that examining these dimensions is important for getting a better understanding of geographic heterogeneity in research “outputs”. By contrast, heterogeneity in research “inputs” is, to our mind, outside of this work’s scope. To be clear, we concur that countries and entire regions draw from a different research endowment when engaging in this rapidly accelerating AI research enterprise. We therefore highlight in our discussion, for example, the marked absence of contributions from Africa and Latin America and call for further research as how to support these areas in joining this research enterprise or, at least, in benefiting from its findings. We thus hope that you can sympathize with our viewpoint, that a “normalization” with respect to financial or human capital is not in the scope of this paper and its intended contribution to the literature.

3. Using JIF as a surrogate for an article’s impact is short-sighted. It is problematic because JIF is a rough indication of a journal quality at best, not individual articles. It is puzzling that why a cutoff of 20 is used to consider high impact for biomedical AI research, because this is also unrealistic as there not many specific journals for this research field with an IF of 20 or higher. For instance, papers published in Nature Communications would not be considered as high impact.

9. We concur that the journal impact factor (JIF) is an imperfect metric for article-level quality. Prior research (including our own) has shown that article-level citations within journal are skewed and that the average number of citations to articles within a journal can deviate from citations an individual article draws. That said, citations require time to accumulate. The JIF provides a more immediate assessment of average quality an article likely needs to exhibit for being published in a journal beyond a certain impact factor threshold. For these reasons, we used the JIF as a proxy for quality, and we agree that it can only serve as a proxy. We used the applied JIF tiering to include the highest-impact field as well as general science journals. We also decided for a threshold of 20 because it splits our sample into approximately the 100 most impactful journals. For reference, we have added a list of these ~100 journals to the supplementary material (S3).

We have also conducted a sensitivity analysis of the JIF categorization, summarizing the share of publications for different JIF tiers by geographic region. As shown in R1.9., this analysis leads to similar conclusions, with higher JIF publications being concentrated in Europe and Northern America, while Asia tends to publish in journals of lower impact.

JIF category	Africa	Asia	Europe	Northern America	Oceania	Latin America	#Articles
<5	1%	38%	27%	29%	2%	2%	120.769
>5 - 10	1%	36%	30%	29%	3%	2%	38.734
>10 - 15	0%	41%	21%	34%	3%	1%	11.745
>15-20	0%	26%	28%	43%	2%	1%	1.955
>20	0%	13%	31%	53%	2%	0%	2.661
							175.864

R1.9. Geography of AI life science research, stratified by JIF categories.

4. In terms of the analysis on the internationally collaboration part, it is incomplete when only first and last author were considered, as they tend to come from the same institute in biomedical research papers. Rather, the authors should examine the affiliation of other co-authors in between. It is also known that more authors tend to lead to more citations – this confounding variable should be adjusted in this analysis. Furthermore, for international collaboration, it would also be interesting to show the co-occurrence information to study which countries prefer to collaborate with each other and its temporal trend.

10. Thank you for this set of constructive suggestions that enabled us to enhance our analyses on international collaborations. We respond to each of the three suggestions – a) interior authors, b) number of authors, c) collaboration patterns – in turn.

11a) To also examine the effect of interior authors in the context of international collaborations, we extended our dataset with the country-specific location information of each author on the article’s author byline, using the *OpenAlex* database for its broader coverage of location. We then designated an article to result from an international collaboration if at least two distinct country locations were present. While our narrower previous definition, considering first and last author only, yielded 20,787 international collaborations (11.82% of the ~175,000 articles), the new definition, considering all listed authors, yields 46,402 international collaborations (26.39%). Given these differences, we have decided to follow your suggestion of considering all authors and updated all analyses accordingly.

11b) The negative binomial regression models on forward citations in dependence of international collaboration showed even larger citation premiums when considering all authors' locations and the numbers of authors on the author byline. We paste Figure 5A included in the main manuscript below (R1.10). We also include the description of the results from the main text here for ease of exposition.

“In the analysis of a potential citation differential between research from international vs. national collaborations, we control for three factors. We include dummy variables for the lead author country to account for country status and regional variance in international collaboration. We control for the number of co-authors because larger author teams are more likely to include a co-author from another country and team size has been shown to correlate with citations. Additionally, we control for the publication year of a focal article to account for the time it had to accrue citations. We find that articles stemming from international collaboration receive, on average, 23% (95% CI 21%, 25%) more citations by general life science articles and 7.5 % (95% CI 2.5%, 12.8%) more citations by clinical life science articles than national collaborations.”

R1.10. Negative binomial regression of forward citations in clinical and general life science research in dependence of international collaborations producing the cited research, accounting for lead author country, number of authors, and publication year.

11c) Lastly, we also extended our analysis of international collaboration patterns. Specifically, we created (i) an alluvial diagram depicting the frequency of international collaborations (R1.11) and (ii) a dynamic visual of the top 10 international collaborations (2000–2022). We describe these results in the main manuscript and briefly below.

R1.11. shows patterns of international collaboration, considering the papers with author bylines for which the lead author's geographic location differs from at least one other author's geographic location. We consider each occurrence of a difference in geographic location separately and sum international collaborations to the regional level. In other words, if a lead author's country is the US and the lead author collaborates with co-authors from China and Germany, then we depict two lines in the alluvial diagram, one from Northern America to Asia and one from Northern America to Europe. The vertical bars on the left depict the sum of outgoing international collaborations from lead authors from the respective region, while the right vertical bars depict the incoming collaborations for non-lead authors from the respective region.

Overall, R1.11. shows, that the European continent engages most frequently in international collaborations, both from an outgoing perspective (lead authors) as well as from an incoming perspective (other authors), represented in the blue vertical bars to both sides of the diagram. European researchers most frequently collaborate with colleagues from the same continent, followed by North American and Asian colleagues. Asian lead authors engage in international collaborations more frequently, than lead authors from Northern America, reflective of the fact that Northern American authors mostly collaborate nationally. Oceania's and Africa's international collaborations appear most pronounced with Asia. Latin America appears more varied in its international collaboration patterns.

R1.11. Alluvial diagram of international collaborations

To further display temporal trends in international collaborations, we have also created a dynamic graph that maps the evolution of dyadic collaborations between continents for the most productive dyads from 2000–2022. We provide the dynamic visual here: <https://public.flourish.studio/visualisation/15308355/>. Following the logic of the alluvial diagram, the continent listed first in the dynamic graph refers to the geographic location of the lead author, the second continent listed refers to the geographic location of a non-lead author. In brief, collaborations between European researchers dominate over time, followed by collaborations between Europe and Northern America. More recently, collaborations among Asian researchers have caught up.

5. The method used to describe Figure 4 should be better explained and described so that readers can fully comprehend its results and findings. For instance, is n in figure 4 the total number of citations? Why is it almost identical for both life science papers vs. clinical AI papers, given there is a significant discrepancy in their quantities in figure 1. Also, what is journal fixed effects? Even after reading the methods part, this is still confusing.

11. We include the revised Figure 4 with a more detailed explanatory caption in the main manuscript.

To elaborate here, we run negative binomial regressions, an econometric model suitable for the distributional properties of our skewed dependent variable, measuring forward citations to a focal paper. The underlying N for both models (i.e., for general forward citations versus forward citations from clinical articles) is the number of articles and remains principally the same, as each focal article is “at risk” of receiving forward citations from both clinical and non-clinical downstream research articles. The slight difference in the underlying N of the two models results from the fact that perfect linear combinations drop out of the regression model. For example, if all papers from a certain journal receive zero citations from clinical articles, then all observations for that journal drop out of the model estimating variance in the likelihood of receiving forward citations from clinical articles. This fact also explains the purpose of the fixed effects, your last question. Fixed effects, here at the journal level, account for unobserved and time-invariant heterogeneity across journals that likely correlates with forward citations, like a journal's status in the scientific community. One can account for this type of heterogeneity by a set of dummy variables for each journal that turn one if a specific article gets published in a specific journal (and zero otherwise). As such, if all articles published in a specific journal never get a citation from a clinical article, then the journal dummy is perfectly linear to the dependent variable of zero citations. In this case, all observations for that specific journal drop from the regression model as there is no remaining within-journal variance in the dependent variable. We hope these detailed explanations address your posed questions in a satisfactory manner.

Other specific comments:

1. The first paragraph should start with a clear definition of AI.

12. We have added a general definition from the Encyclopedia Britannica to the introduction of the main text. We also copy the definition for convenience below (<https://www.britannica.com/technology/artificial-intelligence>).

“AI refers to the ability of a digital computer or computer-controlled robot to perform tasks commonly associated with intelligent beings.”

2. Fig2A and Fig2B: showing a world map seems nice, but this is not informative: this presentation assumes that the readers are familiar with the country's geographical locations. (I don't know if the authors can label all the African country names on this unannotated map.) The map presentation is also biased towards countries of larger sizes. I suggest that the authors should have an annotated map (and remove Antarctica since it is not analyzed) or use other methods to display the country-level statistics. I would prefer a ranked list of all countries for each analyzed statistic.

13. The best visual format we can think of that can depict geographic diversity in AI-research intensity at a global scale continues to be world maps. We agree with you that labeling, at least selective labeling, of countries adds informational value to the exhibit, and we have included revised world maps in Figure 2 of the main manuscript. We also offer country-level statistics in form of tables in the supplementary material **S2**.

3. The authors should clearly describe and compare related studies if this is not the first analysis on such a topic.

14. We have deepened our discussion of related studies in the front section of the manuscript. To further provide a better overview of the literature we identified as relevant to situating our article, we also include a summary of the literature in the supplementary material **S1**.

4. Icite, icite should be iCite

15. We harmonized the language to iCite throughout. Thank you.

5. Impact research should be impactful

16. We adapted the language to impactful research throughout. Thank you.

Reviewer #2 (Remarks to the Author):

This is an interesting analysis that examines the geographic distribution of artificial intelligence research in life science publications. They produce compelling findings about the concentration of AI-based research and citations in the overall literature in the US and China, but a relatively higher distribution of research in other research when accounting for impact factor of journals that publish AI research. Furthermore, they show that international research is more heavily cited in higher-impact venues than national teams. I have several comments:

Introduction:

(1) The first 3 paragraphs could be shortened considerably. An extensive discussion of the possibilities of AI and need for representative training are not needed to get into the crux of the matter, which starts in line 87, that geographic concentration of AI research may not inform care for patients outside of that geography

17. We agree with you and sharpened the introduction to the article. We now start the article by outlining why the geography of AI life science research matters, focusing on (i) the general importance of harnessing global knowledge for scientific progress^{6, 7, 8} and (ii) the heightened risks specific to localized AI life science research due to (a) biased data foundations to the disadvantage of underrepresented populations^{9, 10, 11} and (b) rising barriers to entry for late movers in light of needed investments^{12, 13}. We then summarize the scant yet growing literature on the geography of AI research, that almost exclusively assesses geographic stratification through the dimension of research quantity (i.e., publication counts), contributing to concerns about a US – China “Arms Race” in AI research^{14, 15, 16}.

Against this backdrop, we position our study to contribute by adding two assessment dimensions – quality and relevance – that, when taken together, underscore the global scale of the AI research enterprise as well as the importance of international collaborations for scientific advancement.

Methods

(2) Can the authors better define what they consider “life science” vs. “clinical” research? Does clinical research pertain to phase I, II, III trials? A better definition here would help

18. We apologize for not having been clear about these core terms, and we include a concrete and detailed definition of the respective terms in our revision. In brief:

Considering the life sciences as the “branch of science concerned with the study of living organisms”, we adopt the concrete scope from the National Library of Medicine’s description on the coverage of the *PubMed* database, our core corpus of academic publications for analyses. Accordingly, PubMed covers especially the biomedical sciences as well as the broader life sciences including, for example, bioengineering and behavioral science¹.

To designate an article from this corpus as “clinical”, we drew on the clinical classification employed by Hutchins and colleagues (2019)⁴ that underpins the clinical article indicator in iCite (<https://icite.od.nih.gov>). This classification approach is very specific, essentially classifying clinical articles when the underlying article is a clinical trial. We paste the exact *PubMed* query from the referenced article below that the authors had used to classify clinical articles. We also include this information in the supplementary material (S5). We chose this approach to obtain a precise assessment of the extent to which AI-related research has reached the clinical context.

PubMed query: (("clinical trial"[Publication Type] OR "clinical trial, phase i"[Publication Type] OR "clinical trial, phase ii"[Publication Type] OR "clinical trial, phase iii"[Publication Type] OR "clinical trial, phase iv"[Publication Type]) OR "clinical study"[Publication Type]).

Your comment further encouraged us to expand our analysis of what may constitute a clinically-relevant article. We orient ourselves using the definition for clinical research provided by Ioannidis (2016)⁵, according to which “clinical research is meant to cover all types of investigation that address questions on the treatment, prevention, diagnosis/screening, or prognosis of disease”. To operationalize this definition, we make use of the MeSH (Medical Subject Header) thesaurus to identify all articles in our dataset that get assigned by the National Library of Medicine a disease-signifying MeSH term from the C-branch of the MeSH tree structure (<https://www.nlm.nih.gov/oet/ed/pubmed/mesh/mod01/04-100.html>). We identify about ~43.000 articles in the broader dataset of ~175.000 articles that may be designated as clinically relevant according to this broader definition (i.e., 24,6% of our sample). We offer a table of the most frequent disease MeSH terms in the supplementary material (S5).

We can now compare how sensitive the geography of clinical research is to the operationalization of clinical research. R2.1. lists the 15 most productive countries in terms of AI-focused clinical research, together

¹ <https://pubmed.ncbi.nlm.nih.gov/about/#:~:text=PubMed%20Overview,and%20abstracts%20of%20biomedical%20literature.>

accounting for more than 80% of clinically relevant articles. The ranking of countries and their relative shares of total clinical research productivity are similar for both C-branch MeSH terms (left) as well as iCite definition (right). According to both clinical designations, the US are ahead of other countries, followed by China and the UK. We now offer this sensitivity analysis also in our supplementary material (S5).

	C-branch MeSH term		Clinical (iCite)	
	absolute	relative	absolute	relative
United States	12.495	29%	730	31%
China	7.387	17%	315	13%
United Kingdom	2.249	5%	165	7%
Germany	1.720	4%	147	6%
Japan	1.527	4%	101	4%
South Korea	1.462	3%	84	4%
Italy	1.412	3%	76	3%
Canada	1.397	3%	71	3%
India	1.281	3%	16	1%
France	934	2%	55	2%
Australia	914	2%	48	2%
Spain	877	2%	56	2%
Netherlands	839	2%	73	3%
Taiwan	793	2%	36	2%
Switzerland	491	1%	27	1%
Top 15 countries	35.778	83%	2.000	85%
All countries	43.265	100%	2.345	100%

R2.1. Clinically-relevant research according to C-branch MeSH terms (left) and clinical research indicator from iCite (right), stratified by country

(3) The authors need to acknowledge that their classification of “AI-based research” is prone to error.

a. First, they only identify AI from journal titles or abstracts, which may under-identify AI methods that are key to the paper but only mentioned in the methods.

19. We now include an evaluation of the accuracy with which we identify AI-related life science articles with our employed approach. Following your comment (and the second comment of Reviewer 1), we calculate precision (97%) and recall (93%) statistics for our approach, indicating overall reliable and comprehensive identification of AI life science articles. We still include a text in our limitation section, pointing to different options for identifying especially clinically relevant AI articles.

As to your comment on mining keywords in the titles and abstracts versus full texts, our recall analyses can, at least in part, address this question. For recall, we analyze a set of 148 articles that were part of special issues on AI life science research and find that our identification approach recalls 138 articles (93%). Five of the 148 articles were not recalled due to our employed keywords appearing only in the full text of the article. Your just concern about under-identification due to our keyword text-mining in titles and abstracts versus full articles would, at least according to this analysis, suggest a potential under-identification rate of 3%. Additionally, we would like to point out that searching for AI-related keywords in titles and abstracts increases the likelihood that AI plays an important role in the underlying research. The science community uses titles and abstracts to navigate what research to pay attention to, in turn, incentivizing authors to include the kernel of their research in these special passages of text¹⁷.

b. Second, AI could apply to many things – methodologies used in a retrospective analyses, derivations of a model, prospective validation of AI. Any differentiation between TYPES of research (beyond just “life science” vs. “clinical”) would be very welcome

20. Thank you for this constructive suggestion. Besides the revised analyses with respect to clinical articles presented in our response #19 to your comment above, we followed your suggestion and first stratified the articles in our sample by publication types (e.g., journal article, review etc.; also suggested by Reviewer 1). Second, we also differentiated research according to an approximation of substance matter, using journal categories obtained from the Web of Science (Clarivate Analytics). For example, the journal “Circulation” is assigned to the category “Cardiac and Cardiovascular Systems”, whereas “Nature” is designated as “Multidisciplinary Sciences”. We provide further details on the journal categorization in the Methods and include a list of assigned categories in the supplementary material (S4).

Using this data, we created the two heatmaps depicted in R2.2. and R2.3. Both heatmaps enlist the 30 most productive countries in our data on the x-axis, accounting for 93% of the ~175,000 articles in our dataset. R2.2. ranks the publication types according to their relative frequency across countries, showing that the types “journal article” and “review” account for more than 90% of AI research in the life sciences. We also include this heatmap in **S6** and reference the data in our Discussion.

For the stratification of research by content categories, we focus the heatmap (R2.3) on the 30 most frequent journal categories representing, on average, 74% (standard deviation 4%) of the productivity across the 30 most productive countries in our dataset. The vertical axis of R2.3. ranks the 30 fields according to their relative share in productivity across the countries listed on the horizontal axis. The heatmap indicates that there is an emphasis on the 10 top-ranked fields across countries and world regions. As to more regional research foci, there are a few that may be gleaned from the heatmap, for example, Radiology, Nuclear Medicine and Medical Imaging in Western Europe. Certain areas of research may also stand out at the country level, e.g., Biochemical Research Methods for Israel. Overall, the heatmap elicits global productivity concentration in ten major fields of research with many fields beyond still standing to gain from AI application in the future.

R2.2. and R2.3. Heatmap of relative country-focus with respect to publication types (R2.2.) and publication topics (R2.3). The horizontal axes enlist the top 30 productive countries grouped by continent. The vertical axes depict the underlying publication type (R2.2) publication topics (R2.3), respectively. The color scheme of the heatmaps depict the percentage share of country-specific productivity for a given publication type (R2.2.) and a given publication topic (R2.3.), respectively.

(4) Did the exclusion of “neural network” articles only apply to biological neural networks

21. When mining PubMed indexed articles for “neural net* according to the approach by Liu et al. (2021)³, we noticed in manual quality reviews that we also retrieve articles pertaining to biological neural networks instead of the intended AI methodology. To correct for this unintended identification, we turned to the assigned MeSH (Medical Subject Header) terms of these articles as they are an objective – since externally assigned by the National Library of Medicine – representation of the article’s content. In particular, we created a list of the assigned MeSH terms and manually checked whether any of the assigned MeSH terms were AI-related. We then applied a conservative rule excluding articles that contained the keyword “neural network” in their title or abstract but no other AI-related MeSH terms (4.4% of our sample). A manual inspection of articles that are kept versus excluded from the sample has confirmed the viability of this rule. In conclusion, yes, we noticed the need for and applied this process only to articles on biological neural networks. We now also offer a sample construction chart as Figure 7 in the main manuscript to make this process clearer (also pasted below for convenience, R2.4).

Figure R2.4. Sample construction diagram

(5) Can the authors cite any statistics that the OpenAlex interface accurately classifies authors? What happens if there is an author with the same name as another author?

22. We understand your concern regarding author disambiguation but would like to point out that this concern does not impair our analysis. For the reliability and validity of our results it is critical that we accurately assign papers to countries. This assignment is based on author affiliations as listed on the focal publication and does not require disambiguated author identities. Still, your comment has encouraged us to assess the accuracy of country assignment. Specifically, we hired two independent raters to manually check a random draw of 100 articles from our sample for the correct country assignment. The raters indicated a correct country assignment for 98% of the observations and the observed agreement among raters was 99%. i.e., the raters disagreed in one out of 100 cases. In all, we have no grounds to believe that our data on the geography of AI research would be inaccurate.

(6) It would be interesting to reproduce analyses of AI based research on a per capita basis, e.g. per capita of number of research or per capita of a country’s population. It is somewhat expected that countries with higher populations have more AI articles, just like they would have more non-AI articles.

23. We agree that scientific production is correlated with measures of investment into science, e.g., GDP, %R&D investment of GDP, scientific workforce etc. That said, our manuscript seeks to create a global atlas of contributions to the AI life science research enterprise across three dimensions, assessing the geography of

research with respect to publications (dimension 1), underlying content and quality (dimension 2) and, ultimately, relevance (dimension 3). We submit that examining these dimensions is important for getting a better understanding of geographic heterogeneity in research “outputs”. By contrast, heterogeneity in research “inputs” is, to our mind, outside of this work’s scope. To be clear, we concur that countries and entire regions draw from a different research endowment when engaging in this rapidly accelerating AI research enterprise. We therefore highlight in our discussion, for example, the marked absence of contributions from Africa and Latin America and call for further research as how to support these areas in joining this research enterprise or, at least, in benefiting from its findings. We thus hope that you can sympathize with our viewpoint, that a “normalization” with respect to financial or human capital is not in the scope of this paper and its intended contribution to the literature.

(7) Did the authors collect academic vs. industry-sponsored research? Industry-sponsored research is mentioned in a few places in the manuscript but does not appear to be measured.

24. We apologize for any confusion caused when mentioning industry research. We revised the related text passages (see also our response # 18 to your first comment). The goal of our paper is to analyze the geographic stratification of academic AI research in the life sciences. Therefore, we do not include any industry-specific analyses as part of this revision. We have revised the manuscript (especially the introduction) for making our focus on life science research clear.

Results

(8) How do the authors interpret the lower quality found in Oceania vs. the strong citation premium found?

25. Our analyses of quality (now Figures 2C, 3A, and 3B in the original submission) show that Oceania produces AI life science research of good quality. In particular, the dark blue shading of the Oceanian region in Figure 2C, i.e., Australia, New Zealand, and neighboring countries, underscores this finding. Also Figures 3A and 3B show that the share of high-quality research produced in Oceania (3B) is similar to the share of “lower” quality research published in Oceania (3A). In other words, there is no indication for Oceania to produce disproportionately research of lower quality. Hence, the citation premium found for research from Oceania is in accordance with the higher-quality research produced in this region (Figure 3A and B). This is corroborated by Figure 4, which shows that the unconditional citation premium (Figures 4A and 4B) becomes smaller and insignificant once we condition for the underlying quality (Figures 4C and 4D). We hope our detailed response addresses your question.

Discussion

(9) In the discussion, the statements about inadequate access to AI advancements may be premature. Indeed, taking examples from drug discovery, most clinical trials of any cancer agent, for example, are performed in North American and European populations; however, these innovations could be distributed in other populations. One of the potentials of AI, indeed, is the potential for AI to be distributed more widely than traditional clinical innovations. I believe this article is important enough discussing the geographic underrepresentation of authors in AI research and does not need to stray into patient access to technologies.

26. We agree with you and have adapted our Discussion to point to the importance of future research for examining whether the underrepresentation of certain geographies in the AI research enterprise may correlate with geographic stratification in beneficiaries. We paste an according passage from the Discussion below:

“Our global atlas shows many world regions remain moderately or little involved in the AI life science research enterprise. Less than 5% of global AI research in the life sciences is produced in countries from Africa and Latin America. Notably, these two world regions are home to more than 25% of the world population and experience more than half of the global disease burden. That is not to say that the existing research enterprise does not tackle research questions not germane to these regions. In fact, the possibilities for scaling AI applications across world regions may lead to marked benefits for many countries, even if countries are not all involved to a similar extent in the creation of the research. Still, our findings add to a concern that may be especially applicable in the life sciences. The prowess of AI often depends on the data foundation fed to the learning-based models. If research remains geographically biased, it stands to reason that data foundations become increasingly biased.”

(10) A key limitation is that the authors comment on geographic characteristics of first and last author, but not on the geographic characteristics of the patients studied in the trial. Where there is a correlation there (and the authors may do well to illustrate this correlation from a subsample of their articles), it is possible that North American authors can publish clinical trials of agents from African or Asian nations, often with authors from these nations included as middle authors. This limitation should be made clear.

27. Thank you for highlighting this limitation and also showing us a path to addressing it. Following your suggestion, we manually retrieved the geographic location of the clinical sites reported in the full text of a random sample of 100 clinical studies from our dataset. The site location was available for 99 out of these 100 articles in the full text or from supplementary material. For 90 of these 99 cases (91%) the country of the lead author (last or first) was identical to the country reported as the site of the clinical study. Among the few articles where the countries did not match, we found what you anticipated: the country of the study’s site matched to one of the

countries linked to the interior authors of the study. Nonetheless, we now emphasize that our designation of countries is based on authors and not patients and that the locale of these two groups does not necessarily need to be the same. In line also with our response #27 to your comment above, we call this out in the Discussion as an important area for future research because it points to the possibility that the geography of AI knowledge production is correlated with the geography of benefiting populations. We paste the according Discussion section below:

“Finally, the goal of our study is to provide an atlas of AI-focused life science research. As a corollary, we focused on the “supply-side” of research. Bridging this supply-side perspective to a demand-side perspective seems a fruitful research area, addressing questions like what patient populations stand to gain or lose from AI advances. We offer preliminary analyses suggesting that the geography of the researchers correlates with the geography of patients in clinical trials (S5). Nonetheless, more research is needed to better understand supply-demand dynamics in this context.”

References for point-by-point response to the reviewers

1. Lin Z, Yin Y, Liu L, Wang D. SciSciNet: A large-scale open data lake for the science of science research. *Scientific Data* **10**, 315 (2023).
2. Priem J, Piwowar H, Orr R. OpenAlex: A fully-open index of scholarly works, authors, venues, institutions, and concepts. *arXiv preprint arXiv:220501833*, (2022).
3. Liu N, Shapira P, Yue X. Tracking developments in artificial intelligence research: constructing and applying a new search strategy. *Scientometrics* **126**, 3153-3192 (2021).
4. Hutchins BI, Davis MT, Meseroll RA, Santangelo GM. Predicting translational progress in biomedical research. *PLoS Biology* **17**, e3000416 (2019).
5. Ioannidis JP. Why most clinical research is not useful. *PLoS Medicine* **13**, e1002049 (2016).
6. Jones BF, Wuchty S, Uzzi B. Multi-university research teams: Shifting impact, geography, and stratification in science. *science* **322**, 1259-1262 (2008).
7. Coccia M, Wang L. Evolution and convergence of the patterns of international scientific collaboration. *Proceedings of the National Academy of Sciences* **113**, 2057-2061 (2016).
8. Adams J. The fourth age of research. *Nature* **497**, 557-560 (2013).
9. Ricci Lara MA, Echeveste R, Ferrante E. Addressing fairness in artificial intelligence for medical imaging. *Nature Communications* **13**, 4581 (2022).
10. Seyyed-Kalantari L, Zhang H, McDermott MB, Chen IY, Ghassemi M. Underdiagnosis bias of artificial intelligence algorithms applied to chest radiographs in under-served patient populations. *Nature Medicine* **27**, 2176-2182 (2021).
11. Beam AL, Drazen JM, Kohane IS, Leong T-Y, Manrai AK, Rubin EJ. Artificial intelligence in medicine. Mass Medical Soc (2023).
12. Ahmed N, Wahed M, Thompson NC. The growing influence of industry in AI research. *Science* **379**, 884-886 (2023).
13. Schwalbe N, Wahl B. Artificial intelligence and the future of global health. *The Lancet* **395**, 1579-1586 (2020).
14. Allison G, Schmidt E. *Is China Beating the US to AI Supremacy?* Harvard Kennedy School, Belfer Center for Science and International Affairs (2020).
15. AIShebli B, Cheng E, Waniek M, Jagannathan R, Hernández-Lagos P, Rahwan T. Beijing's central role in global artificial intelligence research. *Scientific reports* **12**, 21461 (2022).
16. Lundvall B-Å, Rikap C. China's catching-up in artificial intelligence seen as a co-evolution of corporate and national innovation systems. *Research Policy* **51**, 104395 (2022).
17. Lerchenmueller MJ, Sorenson O, Jena AB. Gender differences in how scientists present the importance of their research: observational study. *bmj* **367**, (2019).

Reviewers' Comments:

Reviewer #1:

Remarks to the Author:

I thank the authors for responding to my previous comments. However, several key issues remain in study design and as a result, I don't find the results and findings of this work to be robust and in some cases, correct.

1. First and foremost, the document collection for this analysis is not properly constructed. The previously suggested AI/CS conferences are merely a list of examples that publish research relevant to this study. There are many other CS conferences as well as AI conferences for life sciences and medicine (e.g. MICCAI, ACM-BCB, AMIA, etc). The suggestion was for the authors to collect relevant articles from all these conferences, not just on the ones in the short list.

More importantly, given the scope of these conferences, if you look at their entire publications and compare them to PubMed articles, their profile is expected to be different, as you demonstrated in R1.2. The previous comment was for one to identify and select a subset of relevant publications from the entire conference proceedings, rather than blindly including all of them. Simply using keywords may be difficult for identifying relevant results. Automatic text classification would be one of the approaches for this, especially it can complement keyword-based searches.

Liu et al (2021) performed their search on WoS – why is WoS not included as a database in this work?

Measuring recall is generally different and the reported performance is likely to be an overestimate – take the following query as an example, it yields over 10,000 results that are not included in your original query. Considering there are many AI specific algorithms used in biomedicine (SVM is just one example), a significant fraction of relevant studies are potentially missing in this analysis.

```
support vector machines NOT (((((((((((artificial intelligen*[Title/Abstract]) OR (neural net*[Title/Abstract])) OR (machine* learning[Title/Abstract])) OR (expert system*[Title/Abstract])) OR (natural language processing[Title/Abstract])) OR (deep learning[Title/Abstract])) OR (reinforcement learning[Title/Abstract])) OR (learning algorithm*[Title/Abstract])) OR (*supervised learning[Title/Abstract])) OR (intelligent agent*[Title/Abstract])) AND (("2000"[Date - Publication] : "2022/12/31"[Date - Publication]))
```

2. The next major issue related to the identification of publications describing clinical research with AI. It is simply NOT correct to consider research as clinical if it is designated as a clinical trial. As a result, the finding that only 1.5% total articles are in this category is misleading to say the least. Articles examining AI in clinical applications should be considered broadly. For example, why should not such a study be included?

Gulshan V, Peng L, Coram M, Stumpe MC, Wu D, Narayanaswamy A, Venugopalan S, Widner K, Madams T, Cuadros J, Kim R, Raman R, Nelson PC, Mega JL, Webster DR. Development and Validation of a Deep Learning Algorithm for Detection of Diabetic Retinopathy in Retinal Fundus Photographs. JAMA. 2016 Dec 13;316(22):2402-2410. doi: 10.1001/jama.2016.17216. PMID: 27898976.

3. The third issue relates to the newly added experiments with regards to publication type and content analysis.

a. It is well-known that most articles in PubMed are journal articles (as opposed to conference proceeding papers). Therefore, that really should be removed from the overall analysis.

b. To analyze different topics among the set of articles, the authors used the assigned categories of corresponding journals instead of identifying topic themes in the articles themselves. This is problematic because a journal's scope is typically broader and can publish articles of different topics. Even among the articles published by the same journal, their topic themes can be highly different. Additionally, I speculate that the kind of journal categorical topics are not likely to be found interesting and useful by the research community.

As a result, the results of both experiments are not informative nor representative.

4. It was previously commented that JIF is not a good indicator of article's quality by various studies. Yet the authors continue using it as a proxy in this work. Since the authors have used the iCite database for their work, it seems natural they could consider alternative metrics such as Relative Citation Ratio (RCR), which measure the scientific influence of each paper by field- and time-adjusting the citations it has received (<https://icite.od.nih.gov>).

Minor:

iCite is not a database of NLM, as is shown in the link above.

Reviewer #2:

Remarks to the Author:

I believe the authors have answered my main critiques. I do not have substantive additions. Nice work.

Authors' Response to Reviewers on

The Global Geography of Artificial Intelligence in Life Science Research”

Ms. Ref. No.: NCOMMS-23-25307B, Nature Communications

We reproduce the remaining comments of Reviewer 1 in **black**, followed by our responses in **blue**.

Reviewers' comments:

Reviewer #1 (Remarks to the Author):

I thank the authors for responding to my previous comments. However, several key issues remain in study design and as a result, I don't find the results and findings of this work to be robust and in some cases, correct.

1. First and foremost, the document collection for this analysis is not properly constructed. The previously suggested AI/CS conferences are merely a list of examples that publish research relevant to this study. There are many other CS conferences as well as AI conferences for life sciences and medicine (e.g. MICCAI, ACM-BCB, AMIA, etc). The suggestion was for the authors to collect relevant articles from all these conferences, not just on the ones in the short list.

More importantly, given the scope of these conferences, if you look at their entire publications and compare them to PubMed articles, their profile is expected to be different, as you demonstrated in R1.2. The previous comment was for one to identify and select a subset of relevant publications from the entire conference proceedings, rather than blindly including all of them. Simply using keywords may be difficult for identifying relevant results. Automatic text classification would be one of the approaches for this, especially it can complement keyword-based searches.

Liu et al (2021) performed their search on WoS – why is WoS not included as a database in this work?

Measuring recall is generally different and the reported performance is likely to be an overestimate – take the following query as an example, it yields over 10,000 results that are not included in your original query. Considering there are many AI specific algorithms used in biomedicine (SVM is just one example), a significant fraction of relevant studies are potentially missing in this analysis.

```
support vector machines NOT ((((((((((artificial intelligen*[Title/Abstract]) OR (neural net*[Title/Abstract])) OR (machine* learning[Title/Abstract])) OR (expert system*[Title/Abstract])) OR (natural language processing[Title/Abstract])) OR (deep learning[Title/Abstract])) OR (reinforcement learning[Title/Abstract])) OR (learning algorithm*[Title/Abstract])) OR (*supervised learning[Title/Abstract])) OR (intelligent agent*[Title/Abstract])) AND (("2000"[Date - Publication] : "2022/12/31"[Date - Publication]))
```

We acknowledge that our sample creation may not exhaust the entire universe of publications that could theoretically qualify for our analysis. However, we cannot conceive of a single study design that would ensure this to be the case. Nevertheless, we believe that absolute comprehensiveness is not necessary for arriving at representative and valid conclusions, provided there are no systematic biases in the sample creation, ensuring its representativeness for the focus of our study.

Our research has from first submission been distinctly centered on AI-focused life science research, requiring included articles to meet two specific criteria: (i) relevance to the life sciences and (ii) a focus on artificial intelligence. Following your suggestion in the first review round, we studiously explored conference proceedings as a potential source for additional articles. However, upon closer examination, it became evident that the majority of conference publications – in line with your expectation expressed above – do not fulfill the first condition (relevance to the life sciences). We examined a total of 26,372 conference proceedings and identified that 920, i.e., less than 3.5% of these, could be deemed relevant to the life sciences according to a text classifying algorithm, thus becoming potential candidates for inclusion in our sample of 175,000 articles. This analysis further underscores a necessity for self-imposed article selection criteria to determine the relevance of conference proceedings for inclusion in our sample rather than relying on a transparent, reproducible, bibliometric gold-standard for life science articles (i.e., PubMed).

Importantly, even if we were to entertain inclusion of proceedings, the identified 920 publications are associated with countries in almost the same proportion as the publications in our sample. Including them in the analysis would not change the results; there is no systematic bias in the geography between these conference proceedings and the PubMed articles. On these grounds, it remains elusive to us how the reliability and validity of the study's analysis and findings could be called into question.

Regarding the choice of database, we opted for PubMed over Web of Science (WoS) because PubMed is the standard bibliographic reference for life science research. Moreover, the Medline collection, accessible through both PubMed and WoS, is almost identically covered in both databases (Gusenbauer 2022). Using WoS to query Medline publications would not have made a difference.

Lastly, we rely on a peer-reviewed and systematically developed set of keywords to identify AI-focused life science research. This method has been developed and evaluated with respect to the balance between precision and recall (Liu et al. 2021). We apply this method to search for proposed keywords in titles or abstracts, ensuring that AI takes a prominent role in the identified life science articles – another conscious choice in light of the scope of our paper. It is possible that the chosen keywords do not cover the entire universe of AI keywords, but again, these keywords were shown to cover the vast majority of AI applications (Liu et al. 2021), while reducing the likelihood of identifying false positives.

To evaluate the quality of our study design, we constructed recall and precision metrics based on your suggestions from the first review round. Our method reliably identified 93% of articles from special issues of life sciences journals dedicated to Artificial Intelligence. Furthermore, a manual inspection of a random sample of 100 articles confirmed that 97% were consistent with the study's focus. These numbers indicate that while some misclassification exists, it is minimal and does not undermine the generalizability or validity of our results.

Even if we missed 5% (~10,000) of publications that would theoretically qualify for our analysis, the main results of our study would not change. For example, our first analysis shows that the United States is leading AI-focused life science research in terms of quantity, with 46,490 articles published between 2000 and 2022, followed by China with 34,256 articles and the United Kingdom with 9,769 articles. For this order to change, we would need to have systematically missed 12,000 articles published by Chinese authors *only* or over 25,000 articles published by British authors *only*.

In summary, we acknowledge that our sampling approach is not immune to error. However, with your guidance from the first revision round, we were able to explore alternative data sources and establish the comprehensiveness of our sample to the degree that we are confident in the generalizability and validity of our analyses.

References

Gusenbauer, M. (2022). Search where you will find most: Comparing the disciplinary coverage of 56 bibliographic databases. *Scientometrics*, 127(5), 2683-2745.

Liu, N., Shapira, P., & Yue, X. (2021). Tracking developments in artificial intelligence research: constructing and applying a new search strategy. *Scientometrics*, 126(4), 3153-3192.

2. The next major issue related to the identification of publications describing clinical research with AI. It is simply NOT correct to consider research as clinical if it is designated as a clinical trial. As a result, the finding that only 1.5% total articles are in this category is misleading to say the least. Articles examining AI in clinical applications should be considered broadly. For example, why should not such a study be included?

Gulshan V, Peng L, Coram M, Stumpe MC, Wu D, Narayanaswamy A, Venugopalan S, Widner K, Madams T, Cuadros J, Kim R, Raman R, Nelson PC, Mega JL, Webster DR. Development and Validation of a Deep Learning Algorithm for Detection of Diabetic Retinopathy in Retinal Fundus Photographs. *JAMA*. 2016 Dec 13;316(22):2402-2410. doi: 10.1001/jama.2016.17216. PMID: 27898976.

We explicitly state in the revised manuscript that our analysis of clinical relevance is based on clinical trials. Additionally, we offer an additional analysis using C-branch MeSH terms for classification of clinical research, which yields consistent results (see Supplementary Material S5). Under this broader definition, the article you referenced is considered clinical because it encompasses the C-branch MeSH term "Diabetic Retinopathy."

We have included a comparison of the geography of clinical research according to both the narrow and the broad definition in Figure 1 (also found in Supplementary Material S5.2 and Point-by-Point response R1.6). If desired, we can incorporate the analysis using C-branch MeSH terms into the main manuscript and relegate the analysis based on the iCite indicator to the supplementary material.

	C-branch MeSH term		Clinical (iCite)	
	absolute	relative	absolute	relative
United States	12,495	29%	730	31%
China	7,387	17%	315	13%
United Kingdom	2,249	5%	165	7%
Germany	1,720	4%	147	6%
Japan	1,527	4%	101	4%
South Korea	1,462	3%	84	4%
Italy	1,412	3%	76	3%
Canada	1,397	3%	71	3%
India	1,281	3%	16	1%
France	934	2%	55	2%
Australia	914	2%	48	2%
Spain	877	2%	56	2%
Netherlands	839	2%	73	3%
Taiwan	793	2%	36	2%
Switzerland	491	1%	27	1%
Top 15 countries	35,778	83%	2,000	85%
All countries	43,265	100%	2,345	100%

Fig 1. Clinically-relevant research according to C-branch MeSH terms (left) and clinical research indicator from iCite (right), stratified by country

3. The third issue relates to the newly added experiments with regards to publication type and content analysis.

- a. It is well-known that most articles in PubMed are journal articles (as opposed to conference proceeding papers). Therefore, that really should be removed from the overall analysis.
- b. To analyze different topics among the set of articles, the authors used the assigned categories of corresponding journals instead of identifying topic themes in the articles themselves. This is problematic because a journal's scope is typically broader and can publish articles of different topics. Even among the articles published by the same journal, their topic themes can be highly different. Additionally, I speculate that the kind of journal categorical topics are not likely to be found interesting and useful by the research community.

As a result, the results of both experiments are not informative nor representative.

We fully agree with your assessment that the results obtained from the analysis of publication types are not informative, and we therefore did not include these results in the main manuscript. In conducting the analysis of publication types, we heeded the reviewer team's feedback and felt compelled to include the results in the supplement for transparency and completeness.

Regarding the findings by journal category, we contend that the analysis of content using Web of Science (WoS) journal categories is both informative and of interest to the research community. Prior studies with substantial influence in this field, such as Gates et al. 2019, Nielsen et al. 2017, Christie et al. 2020, Dechartres et al. 2017, Wu et al. 2019, and Park et al. 2023, have utilized WoS journal categories to map articles to topic fields and disciplines. However, if there is still a desire for a complementary analysis that designates topics at the article level, we can refer to the article-level concepts provided by the OpenAlex database. An analysis based on this data is presented in the heatmap below (Figure 2).

To create this complementary heatmap, we identified the highest-scoring concept (level 1) for each article in our sample, excluding concepts related solely to artificial intelligence. After assigning each article to one concept, we calculated the share of articles linked to a specific concept for each country. The heatmap depicts these shares for the 30 most productive countries and the 30 most frequent concepts. Bright shading in cells indicates higher shares, while dark shading indicates lower shares.

Like the analysis of content by journal categories, the heatmap reveals that AI-focused life science research predominantly centers around certain fields, such as Neuroscience and Radiology. It also confirms our finding that there is only limited specialization of countries in specific content areas. In summary, both analyses—based on journal-level categories and article-level concepts—present a consistent view. We can include either of the two in the main manuscript.

Fig 2. Heatmap of relative country focus with respect to publication topics designated on the paper level.

References

- Christie, A. P., Abecasis, D., Adjeroud, M., Alonso, J. C., Amano, T., Anton, A., ... & Sutherland, W. J. (2020). Quantifying and addressing the prevalence and bias of study designs in the environmental and social sciences. *Nature communications*, 11(1), 6377.
- Dechartres, A., Trinquart, L., Atal, I., Moher, D., Dickersin, K., Boutron, I., ... & Ravaud, P. (2017). Evolution of poor reporting and inadequate methods over time in 20 920 randomised controlled trials included in Cochrane reviews: research on research study. *Bmj*, 357.
- Gates, A. J., Ke, Q., Varol, O., & Barabási, A. L. (2019). Nature's reach: narrow work has broad impact. *Nature*, 575(7781), 32-34.
- Nielsen, M. W., Andersen, J. P., Schiebinger, L., & Schneider, J. W. (2017). One and a half million medical papers reveal a link between author gender and attention to gender and sex analysis. *Nature human behaviour*, 1(11), 791-796.
- Park, M., Leahey, E., & Funk, R. J. (2023). Papers and patents are becoming less disruptive over time. *Nature*, 613(7942), 138-144.
- Wu, L., Wang, D., & Evans, J. A. (2019). Large teams develop and small teams disrupt science and technology. *Nature*, 566(7744), 378-382.

4. It was previously commented that JIF is not a good indicator of article’s quality by various studies. Yet the authors continue using it as a proxy in this work. Since the authors have used the iCite database for their work, it seems natural they could consider alternative metrics such as Relative Citation Ratio (RCR), which measure the scientific influence of each paper by field- and time-adjusting the citations it has received (<https://icite.od.nih.gov>).

Our atlas is designed to evaluate scientific articles through a multi-dimensional framework, focusing on (1) productivity, (2) adjusted productivity, and (3) relevance. These dimensions offer a comprehensive view of the AI-focused life science research landscape and its geography.

The relevance of an article, measured in citations or adjusted citations as reflected in the Relative Citation Ratio (RCR) provided by iCite, is part of the third dimension – scientific relevance. This dimension is only known ex-post, i.e., after a certain time has passed since publication, allowing for the assessment of the article’s relevance. We considered including the RCR in our assessment of (3) relevance. However, the RCR is strongly correlated with citation counts ($r = 0.7415$, $p < 0.001$), particularly with adjusted citation counts, as shown in our analysis (Figure 4C of the main manuscript). Nevertheless, the RCR would be relevant for dimension (3) and not for dimension (2) of adjusted productivity.

Contrastingly, the Journal Impact Factor (JIF) serves as a forward-looking measure, predicting the expected citations an article will receive based on the journal in which it is published. Although the JIF is not a perfect indicator of quality, it remains the best available metric for estimating an article’s inherent scientific quality at the time of publication. Its widespread use in the research community is well-documented (e.g., Kleppe et al. 2021, Last et al. 2022, Dechartres et al. 2017). We acknowledge and explicitly state the shortcomings of this measure.

We further propose an adjustment to the provided JIF methodology. By evaluating journals based on their JIF ranking within specific fields, rather than solely on their JIF score, we can provide a field-adjusted view of an article's quality. In this adjusted method, the top-ranked journal in each category is categorized as 'exceptional quality.' Our findings from this field-adjusted analysis, presented in Figure 3, closely align with the results based on absolute JIF scores in the main manuscript. The bulk of AI-focused life science research from Asian authors, and its increase over time, is primarily published in journals outside the top-ranked field journals. We have included this analysis below and are open to incorporating these reinforcing analyses into the main manuscript.

Fig 3. Geography of the AI life science research enterprise stratified by six world regions and field-adjusted journal ranking.

References

Dechartres, A., Trinquart, L., Atal, I., Moher, D., Dickersin, K., Boutron, I., ... & Ravaud, P. (2017). Evolution of poor reporting and inadequate methods over time in 20 920 randomised controlled trials included in Cochrane reviews: research on research study. *Bmj*, 357.

Kleppe, A., Skrede, O. J., De Raedt, S., Liestøl, K., Kerr, D. J., & Danielsen, H. E. (2021). Designing deep learning studies in cancer diagnostics. *Nature Reviews Cancer*, 21(3), 199-211.

Last, K., Hübsch, L., Cevik, M., Wolkewitz, M., Müller, S. E., Huttner, A., & Papan, C. (2022). Association between women's authorship and women's editorship in infectious diseases journals: a cross-sectional study. *The Lancet Infectious Diseases*, 22(10), 1455-1464.

Minor:

iCite is not a database of NLM, as is shown in the link above.

Reviewer #2 (Remarks to the Author):

I believe the authors have answered my main critiques. I do not have substantive additions. Nice work.

Reviewers' Comments:

Reviewer #1:

I thank the authors for considering my previous comments. Please find below my point-by-point responses:

1. The collection of papers related to AI and life sciences

“We examined a total of 26,372 conference proceedings and identified that 920, i.e., less than 3.5% of these, could be deemed relevant to the life sciences according to a text classifying algorithm, thus becoming potential candidates for inclusion in our sample of 175,000 articles.” ... “Importantly, even if we were to entertain inclusion of proceedings, the identified 920 publications are associated with countries in almost the same proportion as the publications in our sample.”

First, a conference proceeding is a collection of research papers presented at a specific conference, and it does not refer to an individual paper. For instance, the MICCAI 2023 conference proceeding includes over 700 full papers presented at that conference. Therefore, it is unclear why the proceedings can be switched to “the identified 920 publications.” Furthermore, the details of this text classification experiment are not provided, thus an assessment of its results cannot be done.

“Including them in the analysis would not change the results; there is no systematic bias in the geography between these conference proceedings and the PubMed articles.” “Even if we missed 5% (~10,000) of publications that would theoretically qualify for our analysis, the main results of our study would not change.”

What is the supporting evidence for these claims? There would be at least tens of thousands of papers, if not hundreds of thousands, from conference proceedings that are relevant to both AI and life sciences but are missing in your collection.

“Regarding the choice of database, we opted for PubMed over Web of Science (WoS) because PubMed is the standard bibliographic reference for life science research. Moreover, the Medline collection, accessible through both PubMed and WoS, is almost identically covered in both databases (Gusenbauer 2022). Using WoS to query Medline publications would not have made a difference.”

WoS was previously mentioned because it contains more journals and proceedings outside of the MELDINE collection. This relates to the previous comment that including just publications from PubMed is problematic given the unique interdisciplinary nature of the studied topic, which is at the intersection of computer science/informatics and life sciences.

The SVM example query in my previous comment clearly demonstrated that even limiting search scope to PubMed data only, at least tens of thousands relevant papers would be missed by using the query from Liu et al. Therefore, reporting that your method achieves recall over 90% is questionable. In fact, Liu and colleagues did not claim their method achieves over 90% in recall either.

2. The classification of papers as clinical research

Just by using the disease (C-branch) terms in MeSH, 43,265 articles were classified as clinical research. This represents 24.6% of total publications, in contrast to 1.5% stated in the main manuscript. Thus, it is NOT acceptable to report overly underestimated statistics in the main manuscript while other alternative approaches are available with more reasonable results. The comparison results by relative percentage only show few significant differences (e.g., 13% to 17% for China and 3% vs. 1% for India) but they are also largely affected by how many total papers published in each country.

As a matter of fact, a few research methods have been established for identifying clinical research in PubMed:

<https://pubmed.ncbi.nlm.nih.gov/clinical/>

<https://pubmed.ncbi.nlm.nih.gov/help/#clinical-study-categories-bibliography>

Del Fiol G, Michelson M, Iorio A, Cotoi C, Haynes RB. A Deep Learning Method to Automatically Identify Reports of Scientifically Rigorous Clinical Research from the Biomedical Literature: Comparative Analytic Study. *J Med Internet Res*. 2018 Jun 25;20(6):e10281. doi: 10.2196/10281. PMID: 29941415; PMCID: PMC6037944.

3. Article-level topic analysis

The analysis of content based on article-level concepts – based on my previous suggestion – clearly resulted in more meaningful results in the newly generated Figure 2, compared to its original counterpart in the main manuscript, which is based on journal categories. In the new figure, technical AI research topics such as computer vision, natural language processing, bioinformatics appear in the top 30, along with specific application fields in medicine such as radiology, oncology, etc. There are few topics that remain not intuitive to this reviewer such as optics and nanotechnology.

4. Using JIF as a proxy for selecting high-impact articles

I respectfully disagree that “Journal Impact Factor (JIF) serves as a forward-looking measure, predicting the expected citations an article will receive based on the journal in which it is published.” First, it is a measure that provides a ratio of citations to a journal in a given year to the citable items in the prior two years, rather than predicts future citations. Moreover, it is a measure for a journal rather than for an individual article. Third, using a cut-off threshold of 20 as a proxy for high impact is arbitrary without scientific support. This is concerning because by doing so, it would completely rule out any papers published in highly regarded journals such as *Nature Communications*, *PLoS Medicine*, *Science Translational Medicine* as high impact. Finally, most journals with an IF over 20 between 2000 and 2022 are not in the field of biomedical AI. Hence their IFs say more about their typical content in a specific field than about biomedical AI papers.

The newly proposed adjustment to the raw JIF is less problematic in that while it still relies on JIF, it is at least taking into consideration the differences in research fields. For a long time until recently, almost no journals in the field of (bio-)medical AI have an IF over 10 but this does not mean all papers published in

those journals are not high impact. And this is the unique feature of Relative Citation Ratio (RCR), which contains a key step of field normalization when considering citations on an article level. Hence, I strongly believe RCR is a better metrics to replace JIF in this part of the analysis.

While the authors claim that “Our findings from this field-adjusted analysis, presented in Figure 3, closely align with the results based on absolute JIF scores in the main manuscript.” There are noticeable differences in this new figure vs. the existing one in the main manuscript. For instance, one can see more presence of high-impact research from Latin America and Oceania in the new figure. Also, the overall proportion and trends of high impact research from Asia appears to be noticeably different in those two figures.

5. Incorrect acknowledgment of iCite

I previously pointed out “iCite is not a database of NLM, as is shown in the link above.” The authors did not address this at all and keep stating “by the iCite database of the NLM” in the revised submission. This is simply not correct in fact and does not provide the proper acknowledgement to the NIH Office of Portfolio Analysis (OPA). <https://dpcpsi.nih.gov/opa>

6. New issue (minor)

refs 22 and 34 are the same and are not well formatted.

POINT-BY-POINT RESPONSE

We thank the Reviewer and Editors for the opportunity to revise our manuscript. With the constructive and clear feedback received, we have expanded our data collection and revised the core analyses. In summary:

- Addressing **comment 1**, we increased the sample of AI life science research from 175,865 articles to 397,967 articles (2.2x increase) by
 - Expanding our keyword-based article identification from the 10 keywords proposed by Liu et al. to the 214 keywords advanced by Baruffaldi and colleagues.^{1,2} After identifying Baruffaldi et al. (2020) as the most comprehensive search strategy for AI-related research, we contacted the authors of Baruffaldi et al. to inquire about the applicability of the keywords to the life sciences context. After receiving a positive response from the authors regarding the comprehensiveness and generalizability of the search approach, we adopted the 214 keywords and obtained confirmatory evidence for the appropriateness of the approach in our dataset (see detailed response to comment 1 below).
 - Incorporating 23,465 conference proceedings publications in our analyses, resulting from a full search of over 10,000 conferences indexed in the OpenAlex database, which was recently assessed as the most comprehensive database for non-journal publications.³ Using the search terms of Baruffaldi et al., we obtain 230,000 proceedings publications related to AI, of which ~10% were also related to life sciences.
 - Using the identification of clinical research advanced by Haynes and colleagues (**comment 2**)^{4,5}, we classify about 20% of our sample as clinical research. We also use the “Haynes approach” to find citations to AI-related life science research that result from downstream clinical research, no longer using iCite (**comment 5**).

Our previously reported main findings are confirmed in the expanded dataset, including

- The world regions of Northern America and Asia dominate research production, led by the US and China, which together account for 44% of global AI-related life science articles (2000–2022). The world regions of Africa and Latin America, meanwhile, contribute less than 4% of total AI-related life science research. Across regions we observe little specialization in terms of topics (**comment 3**), suggesting that research foci do not contribute to this bipolar research enterprise in terms of productivity.
- When productivity is adjusted for quality using field-normalized rankings of publishing outlets (**comment 4**), we show that the world regions of Northern America (especially the US), Oceania (especially Australia), and Europe (especially the UK) differ from other world regions in that they publish disproportionately in high-ranked outlets. Thus, the geographic gravity changes as we move from productivity, our first dimension of assessment, to quality-adjusted productivity, our second dimension of assessment in the global atlas.
- Geographic differences in research quality correspond to geographic differences in scientific and clinical relevance, proxied by forward citation counts, highlighting the value of a multidimensional assessment of the research enterprise.
- International collaborations continue to be associated with a citation premium over national collaborations, indicating the importance of overcoming geographically limited research agendas for the advancement of the AI life sciences research enterprise. However, the extent to which geographic areas participate in international collaborations varies widely.

In the following, we submit a comprehensive point-by-point response that addresses all points raised by the Reviewer. We have included the Reviewer comments in black and our response to the comments in blue.

I thank the authors for considering my previous comments. Please find below my point-by-point responses:

1. The collection of papers related to AI and life sciences

“We examined a total of 26,372 conference proceedings and identified that 920, i.e., less than 3.5% of these, could be deemed relevant to the life sciences according to a text classifying algorithm, thus becoming potential candidates for inclusion in our sample of 175,000 articles.” ... “Importantly, even if we were to entertain inclusion of proceedings, the identified 920 publications are associated with countries in almost the same proportion as the publications in our sample.”

First, a conference proceeding is a collection of research papers presented at a specific conference, and it does not refer to an individual paper. For instance, the MICCAI 2023 conference proceeding includes over 700 full papers presented at that conference. Therefore, it is unclear why the proceedings can be switched to “the identified 920 publications.” Furthermore, the details of this text classification experiment are not provided, thus an assessment of its results cannot be done.

“Including them in the analysis would not change the results; there is no systematic bias in the geography between these conference proceedings and the PubMed articles.” “Even if we missed 5% (~10,000) of publications that would theoretically qualify for our analysis, the main results of our study would not change.” What is the supporting evidence for these claims? There would be at least tens of thousands of papers, if not hundreds of thousands, from conference proceedings that are relevant to both AI and life sciences but are missing in your collection.

In response to your concerns, we have greatly expanded the identification of conference proceedings publications. Going beyond the last revision, which examined six conference series, we now consider over ten thousand conferences indexed in OpenAlex. Recent research documents that OpenAlex has the widest coverage of academic publications, especially for non-journal publications.³ Applying the search strategy by Baruffaldi et al., we obtain roughly 230,000 AI-related articles published in conference proceedings (2000–2022).

In a next step, we apply the content classification embedded in the OpenAlex database, the successor to Microsoft Academic Graph (MAG), to identify AI research that also addresses topics relevant to the life sciences. OpenAlex tags articles with multiple concepts representing their topical focus using an automated state-of-the-art machine learning classifier based on titles and abstracts, with confidence scores indicating relevance.⁶ These scientific concepts are organized hierarchically, with 19 root-level concepts branching into six levels of specific topics. When a lower-level concept is mapped, all of its parent concepts are mapped as well, ensuring comprehensive coverage. This structure supports a rich network of interconnected scientific entities, facilitating advanced querying and analysis.⁷ After reviewing pertinent literature, Wang et al 2020 (p. 399) state, for example: “Numerous studies seem to confirm that machine curated results in MAG achieve reasonable if not greater accuracy over commercial data sets with considerable amount of human effort.”⁶ Of note, we used the same content classification algorithm to reclassify the heatmap from journal-level to article-level content during the previously submitted revision (see also comment 3 below).

To identify life science-related articles in conference proceedings, we only consider articles that have been assigned at least one of the following four top level concepts (defined as level 0 in Open Alex) relevant to life science research: Biology, Chemistry, Medicine, or Psychology.

We verified the representativeness of these terms for life science research in our sample of PubMed articles. Here, these four terms represent more than 80% of the indexed research.

We obtain 28,848 conference proceedings publications at the intersection of AI and life sciences, compared to ~900 proceedings publications obtained from the illustrative conferences proposed as part of the last revision. We are able to identify the geography of 23,465 of these articles (~80%). We include a list of the top-ranked conferences in the supplementary material based on rankings provided by CORE, the Computing Research and Education Association of Australasia. CORE provides expert-based assessments of all major conferences in the computing disciplines with information on their research subfield, and is a standard resource for ranking computer science conferences.¹ In addition, we include specific analyses of the sample of conference proceedings publications in the supplementary material.

For the main manuscript, we include this extended set of conference proceedings publications alongside journal publications in all of our analyses. This new data supports our key findings as outlined in our summary response to reviewers and editors above.

“Regarding the choice of database, we opted for PubMed over Web of Science (WoS) because PubMed is the standard bibliographic reference for life science research. Moreover, the Medline collection, accessible through both PubMed and WoS, is almost identically covered in both databases (Gusenbauer 2022). Using WoS to query Medline publications would not have made a difference.”

WoS was previously mentioned because it contains more journals and proceedings outside of the MELDINE collection. This relates to the previous comment that including just publications from PubMed is problematic given the unique interdisciplinary nature of the studied topic, which is at the intersection of computer science/informatics and life sciences. The SVM example query in my previous comment clearly demonstrated that even limiting search scope to PubMed data only, at least tens of thousands relevant papers would be missed by using the query from Liu et al. Therefore, reporting that your method achieves recall over 90% is questionable. In fact, Liu and colleagues did not claim their method achieves over 90% in recall either.

In addition to the expanded search for relevant conference proceedings publications using the OpenAlex database, we also replaced the search strategy proposed by Liu et al. (2021) with the search strategy proposed by Baruffaldi et al (2020). The latter approach represents the most comprehensive search for AI-relevant academic research that we have identified in the literature. For comparison, the approach proposed by Liu et al. (2021) includes 10 keywords for article retrieval versus 214 keywords used by Baruffaldi and colleagues. We include the full list of keywords in the supplementary material. The 20-fold increase in keywords resulted in a 2-fold increase in total articles retrieved, suggesting that the approach by Liu et al (2021) is parsimonious in its identification but less exhaustive than the approach proposed by Baruffaldi and colleagues.

We personally consulted with the authors of Baruffaldi et al (2020) to determine whether the broader list of search terms that the authors had applied across scientific disciplines would be appropriate in the more specific context of the life sciences. Discussions did not result in any concerns about using the keywords to identify AI life science research. We also updated our manual review of the relevance of the identified articles. Two independent raters evaluated a random sample of 150 PubMed articles for AI focus and a random sample of 150 conference articles for life science relevance, increasing our manual inspection of articles by 3x (we had inspected 100 articles as part of our last revision). The inter-rater agreement was 96%, with 90% of PubMed articles rated as having an AI focus and 93% of conference articles rated as being related to the life sciences. Compared to Liu et al. for which we obtained an initial precision estimate of 94%, we achieve a slightly lower level of precision with Baruffaldi et al. However, this trade-off seems acceptable to us given the 2x greater number of articles retrieved in this revision.

Regarding comprehensiveness, we first document that 93% of the articles retrieved by Liu et al. are also retrieved using the identification approach proposed by Baruffaldi et al. In addition, we find that 92% of the articles published in a sample of 15 special issues on AI in life science journals are also part of our sample. Nevertheless, we agree with the reviewer and refrain from reporting specific recall statistics in the manuscript. Instead, we claim that our revision improves the comprehensiveness of our database, as we retrieve about twice as many articles using the broader set of keywords.

We would like to thank the reviewer for pointing out the possibility of a more comprehensive search approach instead of our original approach, which significantly expanded our data base for analysis and also confirmed our findings in this much larger dataset as outlined in our summary response to the reviewer and editors above.

2. The classification of papers as clinical research

Just by using the disease (C-branch) terms in MeSH, 43,265 articles were classified as clinical research. This represents 24.6% of total publications, in contrast to 1.5% stated in the main manuscript. Thus, it is NOT acceptable to report overly underestimated statistics in the main manuscript while other alternatively approaches are available with more reasonable results. The comparison results by relative percentage only show few significant differences (e.g., 13% to 17% for China and 3% vs. 1% for India) but they are also largely affected by how many total papers published in each country. As a matter of fact, a few research methods have been established for identifying clinical research in PubMed:

<https://pubmed.ncbi.nlm.nih.gov/clinical/>

<https://pubmed.ncbi.nlm.nih.gov/help/#clinical-study-categories-bibliography>

Del Fiol G, Michelson M, Iorio A, Cotoi C, Haynes RB. A Deep Learning Method to Automatically Identify Reports of Scientifically Rigorous Clinical Research from the Biomedical Literature: Comparative Analytic Study. *J Med Internet Res.* 2018 Jun 25;20(6):e10281. doi: 10.2196/10281. PMID: 29941415; PMCID: PMC6037944.

Following your recommendation, we applied the approach proposed by Haynes and colleagues to identify clinical research ("Haynes approach")^{4, 5}, replacing the strict iCite classification (iCite essentially designated clinical trials as clinical research). Using the keyword-based Haynes approach, 69,968 (18.7%) of the PubMed articles in our sample were identified as clinical. As expected, conference proceedings had lower rates of clinical concepts embedded in their titles and abstracts (7.4%).

The updated approach to identifying clinical research results in a similar distribution of clinical research productivity to that of total productivity. In particular, the US and China together account for about 45% of clinical and general research productivity (Figure 4 in the main manuscript, blue bars). This finding is similar to the results obtained with the "C-branch" MeSH term identification performed as part of the previous revision. We continue to observe that many countries contribute a 15-20% share of their AI life science productivity to clinical research (Figure 4 in the main manuscript, orange bars). This geographically balanced focus on clinical research also reflects the geographic balance in terms of topic specialization shown in Figure 3 (heatmap) of the main manuscript. For consistency, we now also use the Haynes approach to identify forward citations by clinical research.

3. Article-level topic analysis

The analysis of content based on article-level concepts – based on my previous suggestion – clearly resulted in more meaningful results in the newly generated Figure 2, compared to its original counterpart in the main manuscript, which is based on journal categories. In the new figure, technical AI research topics such as computer vision, natural language processing, bioinformatics appear in the top 30, along with specific application fields in medicine such as

radiology, oncology, etc. There are few topics that remain not intuitive to this reviewer such as optics and nanotechnology.

Given our extended dataset, we also updated the heatmap using the same content classification algorithm used in the previous revision. We obtain consistent results (Figure 3, main manuscript). For example, we observe several technical AI topics ranked highly (computer vision, NLP, speech recognition, information retrieval) alongside medical fields (neuroscience, radiology, surgery, oncology). Optics and nanotechnology also remain on the heatmap with the updated dataset. We conducted expert interviews to test the face validity of the concepts represented in the heatmap. Optics, for example, may emerge because of the increasing use of super-resolution microscopy, which facilitates DNA sequencing, for example. We do not claim that expert interviews can reliably inform validity assessments of nearly 400,000 article-content designations, but at least we did not encounter any categories in the heatmap representation that the experts found implausible. We agree with the reviewer's comment that constructing the heatmap based on article-level content ratings proved very valuable for this part of the analysis, and we have expanded it to include the top 40 concepts across the 40 most productive countries in our expanded dataset.

4. Using JIF as a proxy for selecting high-impact articles I respectfully disagree that "Journal Impact Factor (JIF) serves as a forward-looking measure, predicting the expected citations an article will receive based on the journal in which it is published." First, it is a measure that provides a ratio of citations to a journal in a given year to the citable items in the prior two years, rather than predicts future citations. Moreover, it is a measure for a journal rather than for an individual article. Third, using a cut-off threshold of 20 as a proxy for high impact is arbitrary without scientific support. This is concerning because by doing so, it would completely rule out any papers published in highly regarded journals such as Nature Communications, PLoS Medicine, Science Translational Medicine as high impact. Finally, most journals with an IF over 20 between 2000 and 2022 are not in the field of biomedical AI. Hence their IFs say more about their typical content in a specific field than about biomedical AI papers. The newly proposed adjustment to the raw JIF is less problematic in that while it still relies on JIF, it is at least taking into consideration the differences in research fields. For a long time until recently, almost no journals in the field of (bio-)medical AI have an IF over 10 but this does not mean all papers published in those journals are not high impact. And this is the unique feature of Relative Citation Ratio (RCR), which contains a key step of field normalization when considering citations on an article level. Hence, I strongly believe RCR is a better metrics to replace JIF in this part of the analysis. While the authors claim that "Our findings from this field-adjusted analysis, presented in Figure 3, closely align with the results based on absolute JIF scores in the main manuscript." There are noticeable differences in this new figure vs. the existing one in the main manuscript. For instance, one can see more presence of high-impact research from Latin America and Oceania in the new figure. Also, the overall proportion and trends of high impact research from Asia appears to be noticeably different in those two figures.

We continue to use the adjustment suggested in our last revision, which relies on field-specific rankings of journals and conferences to account for differences in impact across research fields. For journal publications, we consider articles published in the top three journals in the same journal category to be of high quality. For conference proceedings, we consider articles published with an A*-ranked conference according to the CORE ranking.

With the updated dataset, we still observe a larger proportion of AI life sciences research from Northern America and Europe appearing in high-ranked outlets. The previously observed larger contribution of Oceania is also confirmed in the extended dataset (Figure 5, world map in the main manuscript). To better examine the quality distribution of AI life science research within regions, we also include a new figure (Figure 6, main manuscript) that examines the share of publications from a given region appearing in high-ranked outlets over time. This figure shows that in relative proportions all world regions remain fairly stable in their contributions to high-ranked journals.

5. Incorrect acknowledgment of iCite

I previously pointed out “iCite is not a database of NLM, as is shown in the link above.” The authors did not address this at all and keep stating “by the iCite database of the NLM” in the revised submission. This is simply not correct in fact and does not provide the proper acknowledgement to the NIH Office of Portfolio Analysis (OPA). <https://dpcpsi.nih.gov/opa6>.

We no longer use iCite as part of the revised manuscript.

New issue (minor) refs 22 and 34 are the same and are not well formatted.

We have fixed the reference in question. Thank you.

REFERENCES USED IN THE POINT-BY-POINT RESPONSE

1. Baruffaldi S, *et al.* Identifying and measuring developments in artificial intelligence. *OECDiLibrary* (2020).
2. Liu N, Shapira P, Yue X. Tracking developments in artificial intelligence research: constructing and applying a new search strategy. *Scientometrics* **126**, 3153-3192 (2021).
3. Alperin JP, Portenoy J, Demes K, Larivière V, Haustein S. An analysis of the suitability of OpenAlex for bibliometric analyses. *arXiv preprint arXiv:240417663*, (2024).
4. Haynes RB, McKibbin KA, Wilczynski NL, Walter SD, Werre SR. Optimal search strategies for retrieving scientifically strong studies of treatment from Medline: analytical survey. *Bmj* **330**, 1179 (2005).
5. Del Fiol G, Michelson M, Iorio A, Cotoi C, Haynes RB. A deep learning method to automatically identify reports of scientifically rigorous clinical research from the biomedical literature: comparative analytic study. *Journal of medical Internet research* **20**, e10281 (2018).
6. Wang K, Shen Z, Huang C, Wu C-H, Dong Y, Kanakia A. Microsoft academic graph: When experts are not enough. *Quantitative Science Studies* **1**, 396-413 (2020).
7. OpenAlex. Concpets – OpenAlex Technical Documentation (2024).

Reviewers' Comments:

Reviewer #1:

Remarks to the Author:

I'd like to thank the authors for adequately addressing my previous concerns and comments. As a result, the revised manuscript is significantly improved, particularly regarding the completeness of the AI landscape in life sciences and the inclusion of papers relevant to clinical research. I have no further comments, except to suggest that the authors consider adding a brief discussion about the recent emergence of large language models and their implications in this domain. Given the focus on 2000 to 2022 in this paper, it would provide valuable forward-looking context.

Reviewer #3:

None

POINT-BY-POINT RESPONSE

We thank the Reviewer and Editors for the final round of feedback on our manuscript and for accepting our study for publication. Below, we address the remaining comment raised. The reviewer's comment is presented in black, and our response follows in blue:

I'd like to thank the authors for adequately addressing my previous concerns and comments. As a result, the revised manuscript is significantly improved, particularly regarding the completeness of the AI landscape in life sciences and the inclusion of papers relevant to clinical research. I have no further comments, except to suggest that the authors consider adding a brief discussion about the recent emergence of large language models and their implications in this domain. Given the focus on 2000 to 2022 in this paper, it would provide valuable forward-looking context.

Thank you for your detailed and constructive feedback throughout the revision process. We greatly appreciate the time and effort you have invested in reviewing our manuscript. Your comments have significantly helped us improve our work.

Regarding your final comment: we agree that large language models (LLMs) may play an important role in the future of AI-related medical care. We have added a paragraph to our discussion to highlight the relevance of LLMs in the life science context. This addition underscores the potential of LLMs to transform medical applications and the importance of ongoing analysis to capture their evolving impact and ensure equitable benefits across geographies.